# Sweet Gradient Matters: Designing Consistent and Efficient Estimator for Zero-Shot Neural Architecture Search

## Abstract

Neural architecture search (NAS) is one of the core technologies of AutoML for designing high-performance networks. Recently, Zero-Shot NAS has gained growing interest due to its training-free property and super-fast search speed. However, existing Zero-Shot estimators commonly suffer from low consistency, which limits their reliability and applicability. In this paper, we observe that Sweet Gradient of parameters, i.e., the absolute gradient values within a certain interval, brings higher consistency in network performance compared to the overall number of parameters. We further demonstrate a positive correlation between the network depth and the proportion of parameters with sweet gradients in each layer. Based on the analysis, we propose a training-free method to find the Sweet Gradient interval and obtain an estimator, named Sweetimator. Experiments show that Sweetimator has superior consistency compared to existing Zero-Shot estimators in four benchmarks with eight search spaces. Moreover, Sweetimator outperforms state-of-the-art Zero-Shot estimators in NAS-Bench-201 and achieves competitive performance with 2.5x speedup in the DARTS search space.

## 1 Introduction

The computer vision field has witnessed the great success of deep learning. Iconic works such as ResNet (He et al., 2016), MobileNet (Howard et al., 2017; Sandler et al., 2018), and EfficentNet (Tan & Le, 2019) are widely applied for a variety of real-world tasks such as object detection and semantic segmentation. To tackle the trail-and-error shortcomings of handcrafted architectures, Neural Architecture Search (NAS) (Elsken et al., 2019) has been proposed to automatically search powerful networks that even outperform manual designs (Zoph et al., 2018).

A major theme in NAS development is efficiency. From this perspective, NAS can be broadly classified into three categories: All-Shot, One-Shot, and Zero-Shot NAS. All-Shot NAS utilizes approaches such as reinforcement learning (Zoph & Le, 2017) or evolutionary algorithms (Real et al., 2019) to train the sampled architectures one by one during the search process, which costs hundreds or even thousands of GPU days. Based on weight sharing (Pham et al., 2018), One-Shot NAS trains one supernet and utilizes sampling-based (Guo et al., 2020; Chu et al., 2021b; Yu et al., 2020) or gradient-based (Liu et al., 2019; Chen et al., 2019; Xu et al., 2020) approaches, thus reducing the search cost to a few GPU days. Zero-Shot NAS leverages training-free estimators (Mellor et al., 2021; Abdelfattah et al., 2021) to evaluate network performance. As no networks are trained, the search time is reduced to a few GPU hours or even seconds.

However, Zero-Shot NAS commonly suffers from low consistency. Figure 1 illustrates the Spearman's rank between the test accuracy obtained by training the network from scratch and the estimated performance score of mainstream Zero-Shot methods in NAS-Bench-101 (Ying et al., 2019), NAS-Bench-201 (Dong & Yang, 2020; Dong et al., 2022), and NAS-Bench-301 (Zela et al., 2022). The results demonstrate that these methods do not consistently outperform the simple metric of the number of parameters, which limits their reliability and applicability. Moreover, a question is also naturally raised: could we find a Zero-Shot estimator with superior consistency to parameters?

For the networks in the NAS-Bench-101, the NAS-Bench-201, and the NAS-Bench-301 space, we observe that some specific parameters, whose absolute gradient values are in a certain interval, have a

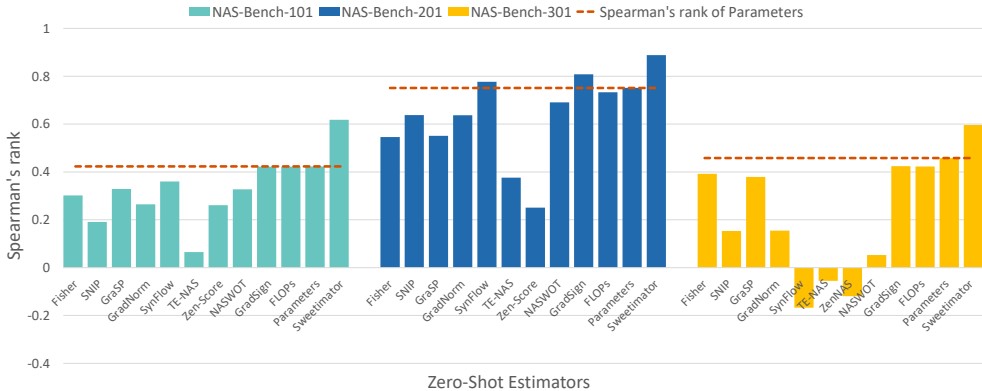

Figure 1: The Spearman's rank correlation coefficient of Zero-Shot estimators on NAS-Bench-101, NAS-Bench-201, and NAS-Bench-301. The dotted line indicates the spearman's rank of Parameters.

stronger consistency with the network performance than the overall number of parameters (Parameters for short). For the sake of brevity, we named the gradient in the such interval as Sweet Gradient. We found an interesting property of Sweet Gradient that the proportion of parameters with Sweet Gradient in each layer is positively correlated with the depth of the network. Based on this property, we propose Sweetimator, an estimator that computes Sweet Gradient interval without training. Figure 1 shows that Sweetimator outperforms the Parameters estimator and achieves the best consistency in all three benchmarks.

The contributions of this work are:

- We observe the Sweet Gradient phenomenon, i.e., the number of parameters with absolute gradient values in a certain interval has better performance consistency than Parameters.
- We demonstrate that there is a positive correlation between the network depth and the proportion of parameters with Sweet Gradient in each layer.
- We propose a simple and effective Zero-Shot estimator, Sweetimator, that can find Sweet Gradient intervals without training.
- In the consistency experiments, Sweetimator outperforms the existing Zero-Shot estimators in four benchmarks with eight search spaces. In the search experiments, Sweetimator has superior performance to state-of-the-art Zero-Shot estimators in NAS-Bench-201 and achieves competitive results with 2.5x speedup in the DARTS search space.

## 2 RELATED WORK

**Neural Architecture Search**. Neural architecture search aims at automatically designing the best-performing network for a specific task. In the early days, Zoph & Le (2017) proposed a reinforcement learning framework to search hyper-parameters of an entire network. Inspired by the modular design paradigm of handcrafted neural networks, NASNet (Zoph et al., 2018) searched cell structures and stacked the searched best normal cell and reduction cell to form a network. Subsequently, Pham et al. (2018) proposed a weight-sharing strategy to reduce the search overhead to a few GPU Days. Afterward, sampling-based approaches (Guo et al., 2020; Chu et al., 2021b; Yu et al., 2020) trained the supernet by path sampling and utilized sub-networks accuracy for evaluation. DARTS (Liu et al., 2019) and its variants (Chen et al., 2019; Xu et al., 2020; Zela et al., 2020; Chu et al., 2021a; Wang et al., 2021; Sun et al., 2022) leveraged differentiable strategies to optimize the supernet and select the final architecture.

**Non-Zero-Shot Estimator**. To facilitate the performance evaluation process, various estimators have been proposed. It is natural to use the validation loss or accuracy (Zoph & Le, 2017; Real et al., 2019; Liu et al., 2018) as a performance estimator. Subsequently, SPOS (Guo et al., 2020) and similar works (Pham et al., 2018; Yu et al., 2020; Chu et al., 2021b) utilize the accuracy of sub-networks as

a proxy for efficient evaluation. Another path for estimators is to utilize machine learning models to predict network performance. NAO (Luo et al., 2018) utilizes an encoder and estimator to find a high-performance network. Wen et al. (2020); Chen et al. (2021c) utilized graph convolutions networks to regress network performance. GP-NAS (Li et al., 2020) proposed a gaussian process based NAS method to obtain the correlations between architectures and performances. TNASP (Lu et al., 2021) uses a transformer and self-evolutionary frameworks to predict network performance. Although the above works can effectively evaluate the network performance, a large search overhead still exists with training networks.

**Zero-Shot Estimator**. Zero-Shot estimators are applied to search architectures without training. NASWOT (Mellor et al., 2021) utilizes activations of rectified linear units to evaluate networks with random initialization. Abdelfattah et al. (2021) proposes a series of zero-cost proxies based on pruning literature (Lee et al., 2019b; Wang et al., 2020; Tanaka et al., 2020). TE-NAS (Chen et al., 2021a) combines neural tangent kernel (NTK) (Lee et al., 2019a) and linear region (Raghu et al., 2017) to evaluate the trainability and expressivity of networks. Zen-NAS (Lin et al., 2021) proposes Zen-Score to measure network performance based on network expressivity. KNAS (Xu et al., 2021) finds that the gradient kernel of the initialized network correlated well with training loss and validation performance. Zhang & Jia (2022) analyzes the sample-wise optimization landscape and proposes GradSign for performance evaluation. A remarkably efficient search is facilitated by the above-mentioned zero-shot estimators. However, compared to network parameters, these methods are not competitive in terms of ranking consistency, raising concerns about their applicability.

## 3 METHOD

### 3.1 PRELIMINARIES

Zero-Shot NAS utilizes network information at initialization for scoring and expects strong rank consistency with the performance. In particular, gradient information is widely adopted in Zero-Shot estimators. For example, SNIP (Abdelfattah et al., 2021) applies gradient values to approximate the change of loss, and GradSign (Zhang & Jia, 2022) calculates the gradient conflict between data samples. Considering that the network parameter is a strong performance estimator as shown in Figure 1, we therefore combine the network parameter with the gradient. Suppose the loss function is $J$, the network parameters are $\theta \in \mathbb{R}^m$ and corresponding average gradients in a mini batch are $\nabla_\theta J \in \mathbb{R}^m$. Then the Zero-Shot estimator to be explored is as follows.

$$Score(thr_1, thr_2) = \sum_{k=1}^{m} \mathbb{I}\left\{ thr_1 \leq \left| \frac{\partial J_n\left(\theta^0\right)}{\partial \theta_k} \right| < thr_2 \right\} \tag{1}$$

where $thr_1, thr_2 \geq 0$ are two thresholds, $\theta^0$ is the network initialization parameter. $\mathbb{I}$ is the indicator function with a value of 1 when the condition is true and 0 otherwise. Equation (1) describes the number of parameters whose absolute gradient values are within a certain interval. $Score(0, +\infty)$ represents the overall network parameters. For the sake of brevity, we utilize $[thr_1, thr_2)$ to describe a interval. We provide a theoretical analysis for the proposed Zero-Shot estimator in Appendix A.

### 3.2 SWEET GRADIENT

We then analyze the rank consistency of Equation (1) with different $[thr_1, thr_2)$ intervals. Figure 2 illustrates the Spearman's rank between the test accuracy of one hundred networks and their scores of Equation (1) on NAS-Bench-101 and NAS-Bench-201. We can see the following patterns:

- The Spearman's rank in the upper right corner of the heatmap can be regarded as the consistency of Parameters which are covered overwhelmingly when $thr_1 = 0$ and $thr_2 = 5$.

- The score with interval $[thr_1, thr_2)$ in the dark blue area yields better rank consistency than Parameters. We name the gradient in such intervals **Sweet Gradient**. The **Sweet Gradient Interval** is defined as: the absolute gradient interval where the number of parameters have better performance consistency than Parameters.

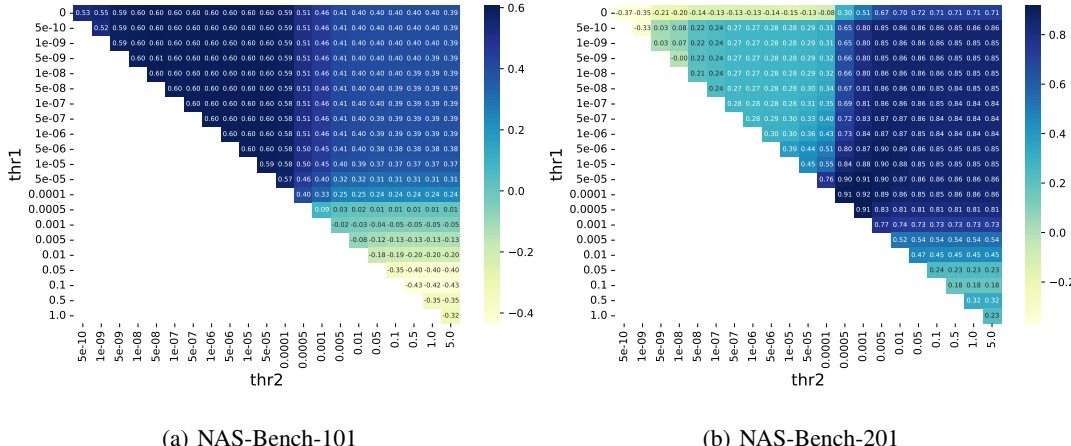

(a) NAS-Bench-101          (b) NAS-Bench-201

Figure 2: The Spearman's rank under different intervals on NAS-Bench-101 and NAS-Bench- 201 (CIFAR10). Each rank is calculated with 100 architectures and a batch size of 128.

Although Figure 2 only shows the Spearman's rank of one hundred architectures in two search spaces, Sweet Gradient exists across different search spaces, datasets, the number of architectures, batch size, and initializations (please refer to Appendix B for more details). This observation indicates that a proper threshold setting can obtain an estimator with better rank consistency than the overall network parameters. However, the thresholds derived based on architecture accuracy still need network training. Does there exist a method to obtain two thresholds without training? This motivates us to investigate the underlying reason for the Sweet Gradient phenomenon.

Reviewing the development of neural networks, depth is one of the most critical factors affecting performance (Simonyan & Zisserman, 2015). Inspired by He et al. (2016); Balduzzi et al. (2017), we dissect the parameters of the different intervals from the perspective of depth. To eliminate the effect of the parameter magnitude, the parameter proportion is used for analysis:

$$Proportion_l = \frac{Score_l(thr_1, thr_2)}{Score_l(0, +\infty)} \tag{2}$$

where $Proportion_l$ is the proportion of parameters with the gradient interval $[thr_1, thr_2)$ in the $l$-th layer, whose range is $[0, 1]$. Figure 3 exhibits the parameter proportion under different layers in NAS-Bench-101 and NAS-Bench-201. More figures can be seen in Appendix C. Surprisingly, the parameter proportion in the Sweet Gradient interval increases as the depth increases, e.g., $[1e-7, 5e-7)$ in NAS-Bench-101 and $[0.0005, 0.001)$ in NAS-Bench-201. In contrast, the parameter proportion in the Non-Sweet Gradient interval decreases with increasing depth or has a large variance, e.g., $[0.01, 0.05)$ in NAS-Bench-101 and $[1e-8, 5e-8)$ in NAS-Bench-201. Another interesting observation is that the parameter proportion tends to increase for small gradients and decrease for large gradients with increasing depth, e.g., $[1e-5, 5e-5)$ and $[0.1, 0.5)$ in NAS-Bench-101.

Therefore, there is a positive correlation between network depth and parameter proportion of Sweet Gradients in each layer. We can seek Sweet Gradient based on the property: *Sweet Gradient is more likely to be located in intervals where the parameter proportion increases as network depth increases*.

Besides the network depth, we also investigated the gradient and activation distribution but found no correlation between the two factors and Sweet Gradient. Please refer to Appendix D for more details.

### 3.3 SWEETIMATOR

**Sweetimator** represents the score in Equation (1) when $[thr_1, thr_2)$ is a Sweet Gradient interval. The discovered property answers the motivational question of how to obtain the Sweet Gradient interval without training, and thus the Sweetimator. Heuristically, we propose sweetness to estimate how sweet the gradient of an interval is:

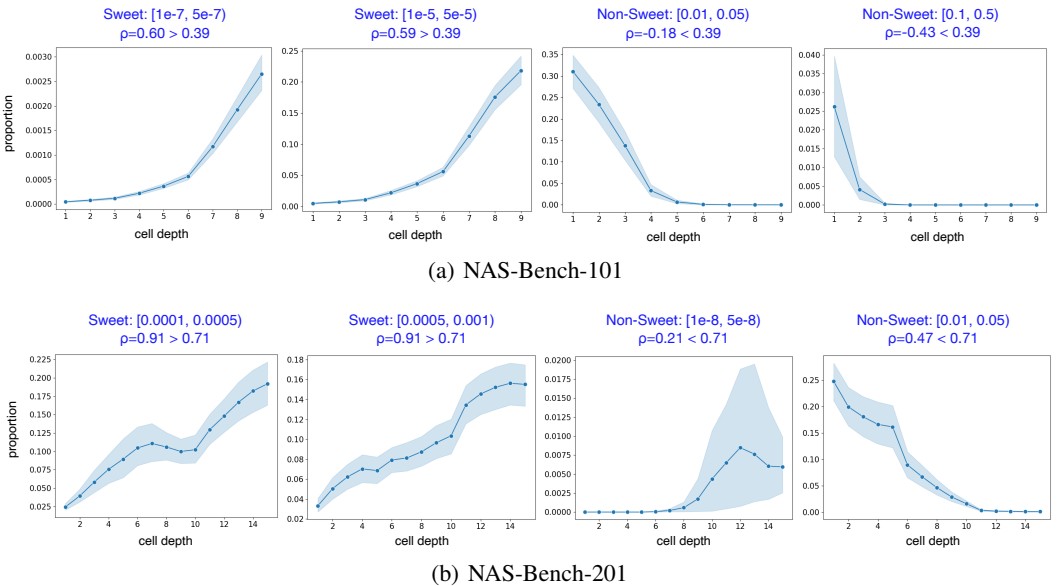

(a) NAS-Bench-101

(b) NAS-Bench-201

Figure 3: The parameter proportion along cell depth of Sweet and Non-Sweet Gradient intervals on NAS-Bench-101 and NAS-Bench-201 (CIFAR-10). Each layer represents a cell of the network and there are 9 and 15 layers in NAS-Bench-101 and NAS-Bench-201, respectively.

$$Sweetness(thr_1, thr_2) = \frac{1}{n} \sum_{k}^{n} (\frac{1}{l_k - 1} \sum_{i}^{l_k - 1} sign(Proportion_{i+1} - Proportion_i)) \quad (3)$$

where $n$ is the number of architectures, $l_k$ denotes the depth of the $k$-th architecture, and the $sign$ function is used to determine whether the parameter proportion is ascending as the depth increases. Equation (3) has a range of $[-1, 1]$, indicating interval incrementality with the depth. $Sweetness = 1$ indicates monotonically ascending and $Sweetness = -1$ indicates monotonically descending. Although both Sweetness and GradSign utilize the $sign$ function, the purposes are completely different. GradSign evaluates the gradient conflict between samples, while we evaluate the interval incrementality for Sweetimator.

In practice, one problem in obtaining the Sweet Gradient is the interval setting, which can be described by the triplet $(max, min, split)$. $max$ means the largest order of magnitude, $min$ means the smallest order of magnitude except zero, and $split$ means each order of magnitude is divided equally into several parts. For example, $(1, 0.1, 2)$ means three interval $[0, 0.1], [0.1, 0.5], [0.5, 1.0]$. We empirically found that $max = 10$ and $min = 1e-10$ are sufficient. And the ablation study of $split$ is provided in Section 4.4. Algorithm 1 describes how to obtain Sweetimator.

---

**Algorithm 1** Sweetimator

---

**Input:** $m$ initialized architectures; $p$ intervals based on $(max, min, split)$;
**Output:** Best Interval for Sweetimator
1: $maxSweetness = -1$
2: $bestInterval = None$
3: **for** $i \leftarrow 1$ to $p$ **do**
4: $\quad Sweetness_i \leftarrow$ Equation (3)
5: $\quad$ **if** $Sweetness_i > maxSweetness$ **then**
6: $\quad\quad maxSweetness = Sweetness_i$
7: $\quad\quad bestInterval = Interval_i$
8: $\quad$ **end if**
9: **end for**
10: **return** $bestInterval$

---

# 4 EXPERIMENT

In this section, we first evaluate the ranking consistency of Sweetimator in NAS-Bench-101 (Ying et al., 2019), NAS-Bench-201 (Dong & Yang, 2020; Dong et al., 2022), NAS-Bench-301 (Zela et al., 2022), and NDS (Radosavovic et al., 2019) benchmarks. Then, we use Sweetimator to conduct search experiments in NAS-Bench-201 and DARTS (Liu et al., 2019) search spaces. Finally, we provide ablation experiments for further analysis. Experimental details can be referred to Appendix E.

## 4.1 CONSISTENCY RESULTS

**Benchmarks**. We conduct experiments in NAS-Bench-101, NAS-Bench-201, NAS-Bench-301, and NDS search spaces. NAS-Bench-101 is the first large-scale NAS Benchmark with 423k architectures and corresponding accuracy on CIFAR-10. NAS-Bench-201 is an extension to NAS-Bench-101, containing 15625 architectures and CIFAR-10, CIFAR-100, and ImageNet16-120 datasets. NAS-Bench-301 is the first surrogate NAS benchmark, with $10^{18}$ architectures on CIFAR-10. NDS statistically analyzes multiple network design spaces, including NAS-Net (Zoph et al., 2018), Amoe-baNet (Real et al., 2019), PNAS (Liu et al., 2018), ENAS (Pham et al., 2018), and DARTS (Liu et al., 2019) search space. And each space has thousands of architectures in the NDS benchmark.

**Baselines**. We compare common Zero-Shot estimators, including SNIP (Lee et al., 2019b), GraSP (Wang et al., 2020), SynFlow (Tanaka et al., 2020), Fisher (Turner et al., 2020), GradNorm (Abdelfat-tah et al., 2021), NASWOT (Mellor et al., 2021), TE-NAS (Chen et al., 2021a), Zen-Score (Lin et al., 2021), GradSign (Zhang & Jia, 2022), and FLOPs and Parameters of networks. A comparison with Non-Zero-Shot estimators is in Appendix F.

**Settings**. The consistency experiments are divided into two groups. The first group contains NAS-Bench-101, NAS-Bench-201, and NAS-Bench-301 with 4500, 15625, and 5000 architectures to test consistency, respectively. For fairness, the batch size of all Zero-Shot estimators is 64. The evaluation metric is Spearman's rank. The second group includes NAS-Net, AmoebaNet, PNAS, ENAS, and DARTS spaces of NDS benchmark with 4846, 4983, 4999, 4999, and 5000 architectures for evaluation, respectively. The architectures are trained on CIFAR-10. All Zero-Shot estimators have a batch size of 128. The evaluation metric is Kendall's Tau.

**Results**. Tables 1 and 2 show that Sweetimator significantly outperforms existing Zero-Shot estimators. For example, Sweetimator is better than the estimator Parameters by 46% (0.423 vs. 0.618) in NAS-Bench-101 in Table 1. The results verify the effectiveness of the proposed method. Moreover, we observe that existing Zero-Shot estimators do not consistently outperform Parameters in terms of rank consistency, which is also found by Ning et al. (2021). In contrast, Sweetimator significantly outperforms Parameters, suggesting that Sweet Gradient is a direction worth exploring for NAS.

Table 1: Rank Consistency of Zero-Shot estimators in NAS-Bench-101, NAS-Bench-201, and NAS-Bench-301 by Spearman's rank.

| Estimators | NAS-Bench-101 | NAS-Bench-201 | | | NAS-Bench-301 |
|---|---|---|---|---|---|
| | CIFAR-10 | CIFAR-10 | CIFAR-100 | ImageNet16-120 | CIFAR-10 |
| Fisher | 0.302 | 0.546 | 0.548 | 0.491 | 0.392 |
| SNIP | 0.191 | 0.638 | 0.637 | 0.578 | 0.153 |
| GraSP | 0.329 | 0.551 | 0.549 | 0.553 | 0.379 |
| GradNorm | 0.265 | 0.637 | 0.637 | 0.578 | 0.155 |
| SynFlow | 0.360 | 0.777 | 0.763 | 0.751 | -0.167 |
| TE-NAS | 0.065 | 0.376 | 0.350 | 0.335 | -0.055 |
| Zen-Score | 0.261 | 0.251 | 0.260 | 0.319 | -0.119 |
| NASWOT | 0.327 | 0.691 | 0.704 | 0.700 | 0.053 |
| GradSign | 0.422 | 0.808 | 0.792 | 0.783 | 0.424 |
| FLOPs | 0.422 | 0.733 | 0.708 | 0.673 | 0.423 |
| Parameters | 0.423 | 0.751 | 0.727 | 0.690 | 0.458 |
| Sweetimator | **0.618** | **0.888** | **0.859** | **0.835** | **0.596** |

Table 2: Rank Consistency of Zero-Shot estimators in five search spaces of NDS by Kendall's Tau.

| Estimators | DARTS | ENAS | PNAS | NASNet | Amoeba |
|---|---|---|---|---|---|
| GradNorm | 0.227 | 0.055 | 0.109 | -0.080 | -0.116 |
| SynFlow | -0.001 | -0.092 | -0.090 | -0.191 | -0.001 |
| NASWOT | 0.480 | 0.387 | 0.363 | 0.299 | 0.208 |
| GradSign | 0.537 | 0.424 | 0.396 | 0.290 | 0.250 |
| FLOPs | 0.500 | 0.413 | 0.395 | 0.288 | 0.238 |
| Parameters | 0.493 | 0.411 | 0.387 | 0.289 | 0.241 |
| Sweetimator | **0.568** | **0.506** | **0.458** | **0.449** | **0.360** |

Table 3: Mean ± std accuracies in NAS-Bench-201. The upper and lower parts indicate results of Architecture Selection and Assisted NAS, respectively. All searches run for 500 times. For a fair comparison, architectures are searched on CIFAR-10 and evaluated on CIFAR-10, CIFAR-100 and ImageNet16-120. Note that we rerun GradSign-assisted NAS experiments using their released code.

| Methods | CIFAR-10 | | CIFAR-100 | | ImageNet16-120 | |
|---|---|---|---|---|---|---|
| | validation | test | validation | test | validation | test |
| *Architecture Selection* | | | | | | |
| Random | 83.20±13.28 | 86.61±13.46 | 60.70±12.55 | 60.83±12.58 | 33.34±9.39 | 33.13±9.66 |
| NASWOT (N=10) | 89.14±1.44 | 92.44±1.13 | 68.50±2.03 | 68.62±2.04 | 41.09±3.97 | 41.31±4.11 |
| Sweetimator (N=10) | **89.53±1.17** | **92.75±0.93** | **68.89±1.97** | **69.02±1.96** | **41.54±3.25** | **41.71±3.34** |
| Optimal (N=10) | 89.92±0.75 | 93.06±0.59 | 69.61±1.21 | 69.76±1.25 | 43.11±1.85 | 43.30±1.87 |
| SynFlow (N=100) | 89.83±0.75 | 93.12±0.52 | 69.89±1.87 | 69.94±1.88 | 41.94±4.13 | 42.26±4.26 |
| NASWOT (N=100) | 89.55±0.89 | 92.91±0.99 | 69.35±1.70 | 69.48±1.70 | 42.81±3.05 | 43.10±3.16 |
| GradSign (N=100) | 89.84±0.61 | 93.31±0.47 | 70.22±1.32 | 70.33±1.28 | 42.07±2.78 | 42.42±2.81 |
| Sweetimator (N=100) | **90.47±0.74** | **93.45±0.57** | **70.40±1.54** | **70.54±1.52** | **43.49±2.10** | **43.71±2.19** |
| Optimal (N=100) | 91.05±0.28 | 93.84±0.23 | 71.45±0.79 | 71.56±0.78 | 45.37±0.61 | 45.67±0.64 |
| NASWOT (N=1000) | 89.69±0.73 | 92.96±0.81 | 69.98±1.22 | 69.86±1.21 | 44.44±2.10 | 43.95±2.05 |
| Sweetimator (N=1000) | **91.00±0.48** | **93.83±0.46** | **71.38±1.32** | **71.62±1.34** | **44.68±1.25** | **45.05±1.45** |
| Optimal (N=1000) | 91.34±0.18 | 94.05±0.22 | 72.15±0.81 | 72.17±0.83 | 45.57±0.73 | 45.79±0.78 |
| *Assisted NAS* | | | | | | |
| REA | 91.08±0.45 | 93.85±0.44 | 71.59±1.33 | 71.64±1.25 | 44.90±1.20 | 45.25±1.41 |
| A-REA | 91.20±0.27 | - | 71.95±0.99 | - | 45.70±1.05 | - |
| G-REA | 91.25±0.57 | 94.10±0.48 | 72.56±1.53 | 72.62±1.46 | 45.62±1.31 | 45.77±1.28 |
| S-REA | **91.43±0.24** | **94.23±0.23** | **72.95±1.07** | **72.96±1.01** | **45.98±0.77** | **46.10±0.69** |
| REINFORCE | 90.32±0.89 | 93.21±0.82 | 70.03±1.75 | 70.14±1.73 | 43.57±2.09 | 43.64±2.24 |
| G-REINFORCE | 90.78±0.68 | 93.61±0.57 | 70.88±1.46 | 71.02±1.36 | 44.53±1.46 | 44.65±1.58 |
| S-REINFORCE | **91.01±0.46** | **93.76±0.46** | **71.36±1.26** | **71.44±1.20** | **44.76±1.22** | **45.02±1.37** |
| BOHB | 91.84±0.49 | 93.64±0.49 | 70.82±1.29 | 70.92±1.26 | 44.36±1.37 | 44.50±1.50 |
| G-BOHB | 91.18±0.29 | 93.96±0.28 | 71.91±0.96 | 71.96±0.89 | 45.32±0.89 | 45.51±0.93 |
| S-BOHB | **91.21±0.22** | **93.98±0.23** | **72.01±0.83** | **72.06±0.76** | **45.38±0.80** | **45.62±0.85** |

## 4.2 SEARCH RESULTS IN NAS-BENCH-201

Following Mellor et al. (2021); Zhang & Jia (2022), we conduct architecture selection and assisted NAS experiments to evaluate the ability of Sweetimator to search high-performance architectures.

**Architecture Selection Settings**. Architecture selection refers to randomly sampling $N$ candidate architectures from the search space and then evaluating the architecture with the largest Sweetimator score. The comparison methods include NASWOT, GradSign, Random and Optimal, where Random uniformly selects architectures from the search space, and Optimal means picking the network with the highest test accuracy among $N$ architectures. Following Mellor et al. (2021), we conduct Sweetimator for sample size of $N = 10$, $N = 100$ and $N = 1000$.

**Sweetimator-Assisted NAS Settings**. In the experiment, the NAS algorithms of REA (Real et al., 2019), REINFORCE (Williams, 1992) and BOHB (Falkner et al., 2018) are assisted with Sweetimator to verify the applicability. For a fair comparison, we utilized the assisted algorithm proposed in GradSign (Zhang & Jia, 2022). In particular, the random selection of each NAS algorithm is replaced by Sweetimator-assisted selection. The NASWOT-, GradSign-, Sweetimator-assisted NAS algorithms are abbreviated as A-, G-, S-algorithm, e.g., S-REA, respectively. Following Zhang & Jia (2022), all results are searched with a time budget of 12000s.

**Results**. Table 3 summarizes the search results. In architecture selection experiments, Sweetimator outperforms compared baselines with a closer distance to Optimal, indicating the ability to search high-performance architectures. In the Sweetimator-assisted NAS experiments, the average accuracy of Sweetimator-assisted was higher than the original algorithm and other estimator-assisted algorithms with lower standard deviation, further validating the efficiency and applicability of Sweetimator.

## 4.3 SEARCH RESULTS IN DARTS SEARCH SPACE

**Settings**. DARTS (Liu et al., 2019) is a popular search space to evaluate NAS algorithms. We conduct experiments on CIFAR-10 (Krizhevsky & Hinton, 2009) and ImageNet (Olga et al., 2015) dataset. In the search phase, we utilize 100 architectures and a batch size of 128 to obtain the best interval of Sweetimator. Further, we integrate Sweetimator with REA algorithm (Real et al., 2019) for searching. The hyper-parameters of REA are followed by Dong & Yang (2020) with 200 cycles. Despite small cycles, the score converges rapidly and the search results can be competitive with other Zero-Shot methods, which also bring a fast search speed. In the retraining phase, we follow DARTS settings to build and train searched network for a fair comparison. More details can be referred to Appendix E.

**Results**. In Table 4, the middle part shows the results on CIFAR-10. Compared to Zero-Shot methods TE-NAS and NASI-ADA, Sweetimator achieves 2.5x speedup (0.01 vs. 0.004) and better performance (2.63 vs. 2.62). When compared to One-Shot methods such as PC-DARTS, Sweetimator has a significantly faster search speed (0.1 vs 0.004). The right part shows the results on ImageNet which also demonstrate that Sweetimator achieves competitive performance and speed compared to other Zero-Shot methods. The searched cells are visualized in Appendix G.

Table 4: Comparison with state-of-the-art NAS methods on CIFAR10 and ImageNet. The Sweetimator results on CIFAR-10 come from four independent searches with different random seeds. Top-1 and Top-5 represent the test error. GDays is GPU days. † Architectures are directly searched on ImageNet. ‡ Time is recorded on one GTX 1080Ti.

| Methods | CIFAR-10 | | | ImageNet | | | |
|---|---|---|---|---|---|---|---|
| | Test Error (%) | Params (M) | GDays | Top-1 (%) | Top-5 (%) | Params (M) | GDays |
| *All-Shot NAS* | | | | | | | |
| NASNet (Zoph et al., 2018) | 2.65 | 3.3 | 2000 | 26.0 | 8.4 | 5.3 | 2000 |
| AmoebaNet (Real et al., 2019) | 3.34±0.06 | 3.2 | 3150 | 24.3 | 7.6 | 6.4 | 3150 |
| PNAS (Liu et al., 2018) | 3.41±0.09 | 3.2 | 225 | 25.8 | 8.1 | 5.1 | 225 |
| *One-Shot NAS* | | | | | | | |
| DARTS (2nd) (Liu et al., 2019) | 2.76±0.09 | 3.3 | 1.0 | 26.7 | 8.7 | 4.7 | 1.0 |
| SNAS (Xie et al., 2019) | 2.85±0.02 | 2.8 | 1.5 | 27.3 | 9.2 | 4.3 | 1.5 |
| GDAS (Dong & Yang, 2019) | 2.82 | 2.5 | 0.17 | 26.0 | 8.5 | 5.3 | 0.21 |
| BayesNAS (Zhou et al., 2019) | 2.81±0.04 | 3.4 | 0.2 | 26.5 | 8.9 | 3.9 | 0.2 |
| P-DARTS (Chen et al., 2019) | 2.5 | 3.4 | 0.3 | 24.4 | 7.4 | 4.9 | 0.3 |
| SDARTS (Chen & Hsieh, 2020) | 2.61±0.02 | 3.3 | 1.3 | 25.2 | 7.8 | 5.4 | 1.3 |
| PC-DARTS (Xu et al., 2020) † | 2.57±0.07 | 3.6 | 0.1 | 24.2 | 7.3 | 5.3 | 3.8 |
| DARTS- (Chu et al., 2021a) † | 2.59±0.08 | 3.5 | 0.4 | 23.8 | 7.0 | 4.9 | 4.5 |
| DrNAS (Chen et al., 2021b) † | 2.54±0.03 | 4.0 | 0.4 | 24.2 | 7.3 | 5.2 | 3.9 |
| $\beta$-DARTS (Ye et al., 2022) | 2.53±0.08 | 3.8 | 0.4 | 24.2 | 7.1 | 5.4 | 0.4 |
| *Zero-Shot NAS* | | | | | | | |
| TE-NAS (Chen et al., 2021a) † | 2.63±0.06 | 3.8 | 0.05 | 24.5 | 7.5 | 5.4 | 0.17 |
| NASI-ADA (Shu et al., 2022) † | 2.90±0.13 | 3.2 | 0.01 | 24.8 | 7.5 | 5.2 | 0.01 |
| Sweetimator † | 2.62±0.05 | 3.4 | 0.004 ‡ | 24.8 | 7.6 | 5.2 | 0.01 ‡ |

## 4.4 ABLATION STUDY

Batch size, architecture numbers, and split of the interval are three major factors that influence the gradient calculation and threshold acquisition of Sweetimator. For further analysis, we conduct ablation studies in NAS-Bench-201, which are shown in Table 5.

**Batch Size**. The batch size is an important hyper-parameter that determines the quality of the gradient. The larger the batch size, the more accurate the gradient. Table 5 demonstrates that Spearman's rank commonly becomes higher with the increasing batch size. Therefore, Sweetimator can be more effective by using a larger batch size.

**Architecture Number**. The architecture number affects threshold acquisition. Table 5 shows that a larger architecture number does not change Spearman's rank. This is because the obtained interval is the same across different architecture numbers. The results suggest that a small number of architectures (e.g., 100) is enough to obtain the best interval.

**Split of Interval**. The split influences the granularity of the interval. In Table 5, the Spearman's ranks are improved on CIFAR-10 ($0.888 \rightarrow 0.913$) and CIFAR-100 ($0.859 \rightarrow 0.887$) but almost kept on ImageNet16-120 ($0.835 \rightarrow 0.836$) when split is larger. Consequently, the interval granularity of different datasets is different, indicating that Sweetimator can be more efficient by designing a flexible interval selection algorithm in the future.

Table 5: Rank Consistency of Zero-Shot estimators in NAS-Bench-201 by Spearman's rank. The baseline has a batch size of $64$, architecture number of $100$, and split of $2$.

| Dataset | Baseline | Batch Size | | | Architecture Number | | | Split | | |
|---|---|---|---|---|---|---|---|---|---|---|
| | | 128 | 256 | 512 | 150 | 200 | 300 | 3 | 4 | 5 |
| CIFAR-10 | 0.888 | 0.908 | 0.917 | 0.921 | 0.888 | 0.888 | 0.888 | 0.913 | 0.910 | 0.913 |
| CIFAR-100 | 0.859 | 0.881 | 0.888 | 0.892 | 0.859 | 0.859 | 0.859 | 0.886 | 0.883 | 0.887 |
| ImageNet16-120 | 0.835 | 0.815 | 0.836 | 0.838 | 0.835 | 0.835 | 0.835 | 0.835 | 0.824 | 0.836 |

## 5 LIMITATIONS AND FUTURE WORK

We only consider computer vision classification tasks and cell-based search spaces in this work. And finding the optimal gradient interval is still a process of discrete selection rather than a continuous differentiable optimization process. In future works, we intend to apply Sweetimator to diverse search spaces (e.g., Transformer space) and explore an optimization-friendly method to obtain the best interval. Besides, it is still a mystery that the Sweet Gradient interval exhibits the property of increasing parameter proportion as network depth increases. We believe a more profound reason exists behind this observation which helps us to better understand deep neural networks. Therefore, a theoretical analysis is necessary in the future work.

## 6 CONCLUSION

This work observes the Sweet Gradient phenomenon that some specific parameters, whose absolute gradient values are in a certain interval, have a stronger consistency with the network performance than Parameters. To obtain the Sweet Gradient interval without training, we investigated the relationship between network depth and Sweet Gradient. We found that Sweet Gradient tends to exist in intervals with increasing parameter proportion as network depth increases. Based on the property, we utilize Sweetness to obtain the best interval. Experiments demonstrate that Sweetimator can achieve superior rank consistency and excellent search results, verifying the effectiveness of the proposed estimator.

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

# A  THEORETICAL ANALYSIS

The dataset is denoted as $\{x_i, y_i\}_{i=1}^N$, where the input $x_i \in \mathbb{R}^d$ and the output $y_i \in \mathbb{R}$. The loss function is

$$J_N(\theta) = \sum_{i=1}^N \ell\left(y_i, f(\theta; x_i)\right) \tag{4}$$

where $f(\theta; x)$ is the network architecture, $\theta \in \mathbb{R}^m$ and $\ell(\cdot, \cdot)$ is the loss function. Then the optimal network parameter $\theta^*$ is

$$\theta^* \triangleq \underset{\theta \in \mathbb{R}^m}{\operatorname{argmin}} J_N(\theta). \tag{5}$$

**Score index**

$$\sum_{k=1}^m \mathbb{I}\left\{\tau_1 \le \left|\frac{\partial J_N\left(\theta^0\right)}{\partial \theta_k}\right| < \tau_2\right\} \tag{6}$$

where $\tau_2 > \tau_1 > 0$ and $\theta^0$ is the initialization parameter. Here $\tau_1$ and $\tau_2$ correspond to $thr_1$ and $thr_2$ in (1).

Regarding the optimal parameter $\theta^*$ and the initial parameter $\theta^0$, we provide theoretical analysis from two perspectives.

- Higher performance. The small loss at the optimal parameter $J_N(\theta^*)$ implies the high network performance. We analyze the upper bound of $J_N(\theta^*)$ and expect it as small as possible;

- Easier to optimize: The small distance between the initial parameter and the optimal parameter means the easy optimization process. We analyze the upper bound of $\left\|\theta^0 - \theta^*\right\|_2$ and expect it as small as possible.

## A.1  THEORETICAL RESULTS

The theorems in (Allen-Zhu et al., 2019) reveal that the neighborhood around random initialization has excellent properties that are almost convex and semi-smooth. Based on the work, we introduce a slightly stronger assumption here. Denote the vector $\ell_2$ norm as $\|\cdot\|_2$.

**Assumption 1** $J_N(\theta)$ is $h$-strong convex and $H$-smooth in the neighborhood $\Gamma\left(\theta^0, R\right) \triangleq \left\{\theta \,\middle|\, \left\|\theta - \theta^0\right\|_2 \le R\right\}$ of the initialization $\theta^0$, where $R > 0$ is the neighborhood radius.

**Lemma 1** $J_N(\theta)$ is $h$-strong convex and $H$-smooth which is equivalent to $h \cdot I \preceq \nabla^2 J_N(\theta) \preceq H \cdot I$, $\forall \theta \in \Gamma\left(\theta^0, R\right)$.

**Theorem 1** *(The upper bound of the loss in the optimal point.) Assume that $J_N(\theta)$ is $h$-strong convex in the neighborhood $\Gamma\left(\theta^0, R\right)$ which in Assumption 1 hold. Then*

$$J_N\left(\theta^*\right) \le J_N(\theta^0) - \frac{1}{2H}\varepsilon^2 m - \frac{1}{2H}\left(\tau_1^2 - \varepsilon^2\right) \cdot \sum_{k=1}^m \mathbb{I}\left\{\left|\frac{\partial J_N\left(\theta^0\right)}{\partial \theta_k}\right| \ge \tau_1\right\} \tag{7}$$

*where $0 \le \varepsilon \triangleq \min\left\{\left|\frac{\partial J_N(\theta^0)}{\partial \theta_k}\right| \,\middle|\, \left|\frac{\partial J_N(\theta^0)}{\partial \theta_k}\right| < \tau_1\right\} < \tau_1$.*

**Remark 1** *The upper bound of $J_N(\theta^*)$ is influenced by two items of $J_N(\theta^0)$ and $\sum_{k=1}^m \mathbb{I}\left\{\left|\frac{\partial J_N(\theta^0)}{\partial \theta_k}\right| \ge \tau_1\right\}$. Considering network performance ranking, we can ignore the effect of the first item since $J_N(\theta^0)$ of different networks is similar. In practice, the network initialization*

*(Glorot & Bengio, 2010; He et al., 2015) tends to maintain the mean and variance of the output in each layer, so the final output logits of different networks have similar distribution, resulting in similar losses. In NAS-Bench-201, the loss of $J_N(\theta^0)$ of all candidate networks on CIFAR10, CIFAR100 and ImageNet16-120 are $2.343 \pm 0.043$, $4.651 \pm 0.042$, and $4.842 \pm 0.047$, respectively. Small deviation indicates that the $J_N(\theta^0)$ of different networks is indeed similar. Therefore, we only need to focus on the second item, i.e, the larger $\sum_{k=1}^{m} \mathbb{I}\left\{\left|\frac{\partial J_N(\theta^0)}{\partial \theta_k}\right| \geq \tau_1\right\}$, the smaller the upper bound of $J_N(\theta^*)$.*

**Theorem 2** *(The upper bound of distance between the initial point and the optimal point.) Assume $J_N(\theta)$ is $H$-smooth in the neighborhood $\Gamma\left(\theta^0, R\right)$ in Assumption 1 hold. Then*

$$\left\|\theta^0 - \theta^*\right\|_2 \leq \frac{1}{h} \cdot \left(M \cdot m - (M - \tau_2) \sum_{k=1}^{m} \mathbb{I}\left\{\left|\frac{\partial J_N\left(\theta^0\right)}{\partial \theta_k}\right| < \tau_2\right\}\right)$$

*where $M \triangleq \max\left\{\left|\frac{\partial J_N(\theta^0)}{\partial \theta_k}\right| \middle| \left|\frac{\partial J_N(\theta^0)}{\partial \theta_k}\right| \geq \tau_2\right\} > \tau_2 > \tau_1 > 0.$*

**Remark 2** *To make the upper bound of $\left\|\theta^0 - \theta^*\right\|_2$ small, the index $\sum_{k=1}^{m} \mathbb{I}\left\{\left|\frac{\partial J_N(\theta^0)}{\partial \theta_k}\right| < \tau_2\right\}$ need to be as large as possible.*

For given $\tau_2 > \tau_1 > 0$, considering the two goals of high performance and easy optimization, we expect the upper bound of $J_N(\theta^*)$ and $\left\|\theta^0 - \theta^*\right\|_2$ to be as small as possible. Combining Theorem 1 and 2, the index

$$\sum_{k=1}^{m} \mathbb{I}\left\{\left|\frac{\partial J_N\left(\theta^0\right)}{\partial \theta_k}\right| \geq \tau_1\right\} \cdot \mathbb{I}\left\{\left|\frac{\partial J_N\left(\theta^0\right)}{\partial \theta_k}\right| < \tau_2\right\} = \sum_{k=1}^{m} \mathbb{I}\left\{\tau_1 \leq \left|\frac{\partial J_N\left(\theta^0\right)}{\partial \theta_k}\right| < \tau_2\right\}$$

need to be as large as possible.

According to Theorems 1 and 2, we can obtain the proposed Zero-Shot estimator.

## A.2 PROOF OF LEMMA 1

Let's just simplify $J_N(\theta)$ to $J(N)$ for the sake of the statement. Denote $J(\theta), \forall \theta \in \Gamma\left(\theta^0, R\right)$ is $h$-strong convex and $H$-smooth as "**Left**" and $h \cdot I \preceq \nabla^2 J_N(\theta) \preceq H \cdot I, \forall \theta \in \Gamma\left(\theta^0, R\right)$ as "**Right**" respectively.

**Left $\Rightarrow$ Right.**
Since $J(\theta)$ is $h$-strong convex, $F(\theta) = J(\theta) - \frac{1}{h}\|\theta\|_2^2$ is convex. Then for the convex function $F(\theta)$, we have

$$\left(\nabla F\left(\theta_1\right) - \nabla F\left(\theta_2\right)\right)^\top \left(\theta_1 - \theta_2\right) \geq 0, \forall \theta_1, \theta_2 \in \Gamma\left(\theta^0, R\right). \tag{8}$$

Notice that $\nabla F\left(\theta\right) = \nabla J\left(\theta\right) - h\theta$ and substitute it into Equation (8). Then

$$\left\|\nabla J\left(\theta_1\right) - \nabla J\left(\theta_2\right)\right\|_2 \left\|\theta_1 - \theta_2\right\|_2 \geq \left(\nabla J\left(\theta_1\right) - \nabla J\left(\theta_2\right)\right)^\top \left(\theta_1 - \theta_2\right) \geq h \left\|\theta_1 - \theta_2\right\|_2^2 \tag{9}$$

where the first equality comes from Cauchy-schwartz inequality.
Since $J(\theta)$ is $L$-smooth, $J(\theta)$ is Lipschitz continuous and the Lipschitz constant is $H$, i.e., $\left\|\nabla J(\theta_1) - J(\theta_2)\right\|_2 \leq H \left\|\theta_1 - \theta_2\right\|_2, \forall \theta_1, \theta_2 \in \Gamma\left(\theta^0, R\right)$. By Cauchy-schwartz inequality, we can obtain that

$$\left(\nabla J\left(\theta_1\right) - \nabla J\left(\theta_2\right)\right)^\top \left(\theta_1 - \theta_2\right) \leq \left\|\nabla J\left(\theta_1\right) - \nabla J\left(\theta_2\right)\right\|_2 \left\|\theta_1 - \theta_2\right\|_2 \leq H \left\|\theta_1 - \theta_2\right\|_2^2. \tag{10}$$

Combing Equation (9) and (10), we have

$$h \left\|\theta_1 - \theta_2\right\|_2 \leq \left\|\nabla J\left(\theta_2 + tv\right) - \nabla J\left(\theta_2\right)\right\| \leq H \left\|\theta_1 - \theta_2\right\|_2. \tag{11}$$

Inspired by Theorem 5.12 in (Beck, 2017), assume $\theta_1 = \theta_2 + tv$ where $t > 0$, then

$$\nabla J\left(\theta_2 + tv\right) - \nabla J\left(\theta_2\right) = \int_0^t \nabla^2 J(\theta_2 + zv)v \, dz \tag{12}$$

Thus

$$ht\|v\|_2 \le \left\| \int_0^t \nabla^2 J(\theta_2 + zv)dz \cdot v \right\|_2 = \|\nabla J(\theta_2 + tv) - \nabla J(\theta_2)\|_2 \le Ht\|v\|_2 \quad (13)$$

Divide both sides by $t$ and let $t \to 0^+$. Then

$$h\|v\|_2 \le \left\| \nabla^2 J(\theta_2) \cdot v \right\|_2 \le H\|v\|_2 \quad (14)$$

By the arbitrariness of $\theta_1$ and $\theta_2$, we have

$$h \cdot I \preceq \nabla^2 J(\theta) \preceq H \cdot I. \quad (15)$$

**Right $\Rightarrow$ Left.**
For $\forall \theta_1, \theta_2 \in \Gamma\left(\theta^0, R\right)$, take $J(\theta)$ out of the Taylor expansion at $\theta_2$ and substitute $\theta_1$ into $J(\theta)$.

$$J(\theta_1) = J(\theta_2) + (\nabla J(\theta_2))^\top (\theta_1 - \theta_2) + \frac{1}{2}(\theta_1 - \theta_2)^\top \nabla^2 J\left(\beta\theta_1 + (1 - \beta)\theta_2\right)(\theta_1 - \theta_2) \quad (16)$$

where $0 < \beta < 1$. By $h \cdot I \preceq \nabla^2 J(\theta) \preceq H \cdot I$, we have

$$J(\theta_1) \ge J(\theta_2) + (\nabla J(\theta_2))^\top (\theta_1 - \theta_2) + \frac{1}{2}h \|\theta_1 - \theta_2\|_2^2 \quad (17)$$

and

$$J(\theta_1) \le J(\theta_2) + (\nabla J(\theta_2))^\top (\theta_1 - \theta_2) + \frac{1}{2}H \|\theta_1 - \theta_2\|_2^2. \quad (18)$$

By the arbitrariness of $\theta_1, \theta_2$, the following conclusion can be obtained by exchanging the position of $\theta_1$ and $\theta_2$

$$J(\theta_2) \ge J(\theta_1) + (\nabla J(\theta_1))^\top (\theta_2 - \theta_1) + \frac{1}{2}h \|\theta_1 - \theta_2\|_2^2 \quad (19)$$

and

$$J(\theta_2) \le J(\theta_1) + (\nabla J(\theta_1))^\top (\theta_2 - \theta_1) + \frac{1}{2}H \|\theta_1 - \theta_2\|_2^2. \quad (20)$$

Adding Equation (17) to Equation (19), we can get

$$(\nabla J(\theta_2) - \nabla J(\theta_1))^\top (\theta_2 - \theta_1) \ge h \|\theta_1 - \theta_2\|_2^2 \quad (21)$$

Similarly, by adding Equation (18) and (20), we have

$$(\nabla J(\theta_2) - \nabla J(\theta_1))^\top (\theta_2 - \theta_1) \le H \|\theta_1 - \theta_2\|_2^2. \quad (22)$$

Thus $J_N(\theta)$ is $h$-strong convex and $H$-smooth.

### A.3   PROOF OF THEOREM 1: THE UPPER BOUND OF THE LOSS IN THE OPTIMAL PARAMETER

Take the loss function $J_N(\theta)$ out of the Taylor expansion at the initialization parameter $\theta^0$. For $\forall \theta \in \Gamma\left(\theta^0, R\right)$, we can obtain that

$$J_N(\theta) = J_N\left(\theta^0\right) + \left[\nabla J_N\left(\theta^0\right)\right]^\top \left(\theta - \theta^0\right) + \frac{1}{2}\left(\theta - \theta^0\right)^\top \nabla^2 J_N\left(\alpha\theta + (1 - \alpha)\left(\theta - \theta^0\right)\right)\left(\theta - \theta^0\right)$$

$$\le J_N\left(\theta^0\right) + \left[\nabla J_N\left(\theta^0\right)\right]^\top \left(\theta - \theta^0\right) + \frac{1}{2}H \cdot \left\|\theta - \theta^0\right\|_2^2$$

$$= J_N\left(\theta^0\right) + \sum_{k=1}^m \frac{\partial J_N\left(\theta^0\right)}{\partial \theta_k} \cdot \left(\theta_k - \theta_k^0\right) + \frac{1}{2}H \cdot \sum_{k=1}^m \left(\theta_k - \theta_k^0\right)^2 \triangleq L_N(\theta) \quad (23)$$

where $0 < \alpha < 1$.
Denote $\overline{\theta}^* \triangleq \underset{\theta \in \Gamma(\theta^0, R)}{\operatorname{argmin}} L_N(\theta)$. We assert that

$$J_N(\theta^*) \le L_N\left(\overline{\theta}^*\right). \quad (24)$$

Otherwise if $J_N(\theta^*) > L_N(\overline{\theta}^*)$, we have

$$J_N\left(\overline{\theta}^*\right) \geq J_N\left(\theta^*\right) > L_N\left(\overline{\theta}^*\right) \tag{25}$$

which is in contradiction with Equation (23).
Notice that $L_N(\theta)$ is a quadratic function of $\theta - \theta^0$. To make the upper bound of $J(\theta^*)$ to be as small as possible, we expect $L_N(\theta)$ to be the global minimum in $\Gamma\left(\theta^0, R\right)$. The minimum point of $L_N(\theta)$ is

$$\overline{\theta}_k^* - \theta_k^0 = -\frac{1}{H}\frac{\partial J_N\left(\theta^0\right)}{\partial \theta_k}, \; k \in \{1, \cdots, m\} \tag{26}$$

Next we prove that the optimal point of $L_N(\theta)$ is in the feasible domain $\Gamma\left(\theta^0, R\right)$.
Since $\theta^*$ is the optimal point of $J_N(\theta)$ and $J_N(\theta)$ is $H$-smooth, we have

$$\left\|\nabla J_N\left(\theta^0\right)\right\|_2 = \left\|\nabla J_N\left(\theta^0\right) - \nabla J_N\left(\theta^*\right)\right\|_2 \leq H\|\theta^* - \theta^0\|_2 \leq HR. \tag{27}$$

Then

$$\left\|\overline{\theta}^* - \theta^0\right\|_2 = \frac{1}{H}\left\|\nabla J_N\left(\theta^0\right)\right\|_2 \leq R, \tag{28}$$

further we have $\overline{\theta}^* \in \Gamma\left(\theta^0, R\right)$.
Thus the global minimum of $L_N(\theta)$ is

$$L_N\left(\overline{\theta}^*\right) = J_N\left(\theta^0\right) - \frac{1}{2H}\sum_{k=1}^m\left(\frac{\partial J_N\left(\theta^0\right)}{\partial \theta_k}\right)^2. \tag{29}$$

For the upper bound of $J_N(\theta^*)$, we can obtain that

$$
\begin{aligned}
J_N\left(\theta^*\right) \leq L_N\left(\overline{\theta}^*\right) &= J_N\left(\theta^0\right) - \frac{1}{2H}\sum_{k=1}^m\left|\frac{\partial J_N\left(\theta^0\right)}{\partial \theta_k}\right|^2 \\
&\leq J_N\left(\theta^0\right) - \frac{1}{2H}\tau_1^2\cdot\sum_{k=1}^m\mathbb{I}\left\{\left|\frac{\partial J_N\left(\theta^0\right)}{\partial \theta_k}\right| \geq \tau_1\right\} - \frac{1}{2H}\cdot\varepsilon^2\cdot\sum_{k=1}^m\mathbb{I}\left\{\left|\frac{\partial J_N\left(\theta^0\right)}{\partial \theta k}\right| < \tau_1\right\} \\
&= J_N(\theta^0) - \frac{1}{2H}\varepsilon^2 m - \frac{1}{2H}\left(\tau_1^2 - \varepsilon^2\right)\cdot\sum_{k=1}^m\mathbb{I}\left\{\left|\frac{\partial J_N\left(\theta^0\right)}{\partial \theta k}\right| \geq \tau_1\right\}
\end{aligned} \tag{30}
$$

where $0 \leq \varepsilon \triangleq \min\left\{\left|\frac{\partial J_N(\theta^0)}{\partial \theta_k}\right|\;\Big|\;\left|\frac{\partial J_N(\theta^0)}{\partial \theta_k}\right| < \tau_1\right\} < \tau_1$.

## A.4 Proof of Theorem 2: the upper bound of distance between the initial parameter and the optimal parameter

Denote the vector $\ell_1$ norm as $\|\cdot\|_1$. Take the gradient of loss function $\nabla J_N(\theta)$ out of the Taylor expansion at the optimal parameter $\theta^*$ and substitute the initialization parameter $\theta^0$ into the gradient of loss function.

$$
\begin{aligned}
\nabla J_N\left(\theta^0\right) &= \nabla J_N\left(\theta^*\right) + \nabla^2 J_N\left(\gamma\theta^0 + (1-\gamma)\theta^*\right)\cdot\left(\theta^0 - \theta^*\right) \\
&= \nabla^2 J_N\left(\gamma\theta^0 + (1-\gamma)\theta^*\right)\cdot\left(\theta^0 - \theta^*\right)
\end{aligned} \tag{31}
$$

where $0 < \gamma < 1$ and the last equality comes from $\nabla J_N\left(\theta^*\right) = 0$. By Assumption 1, we have $h\cdot I \preceq \nabla^2 J_N\left(\gamma\theta^0 + (1-\gamma)\theta^*\right) \preceq H\cdot I$. Thus $h\cdot\left\|\theta^0 - \theta^*\right\|_2 \leq \left\|\nabla J_N\left(\theta^0\right)\right\|_2 \leq H\cdot\left\|\theta^0 - \theta^*\right\|_2$.

Further we can obtain that

$$
\begin{aligned}
\left\|\theta^0 - \theta^*\right\|_2 &\leq \frac{1}{h}\left\|\nabla J_N\left(\theta^0\right)\right\|_2 \leq \frac{1}{h} \cdot \left\|\nabla J_N\left(\theta^0\right)\right\|_1 = \frac{1}{h} \cdot \sum_{k=1}^{m}\left|\frac{\partial J_N\left(\theta^0\right)}{\partial \theta_k}\right| \\
&\leq \frac{1}{h} \cdot \tau_2 \cdot \sum_{k=1}^{m} \mathbb{I}\left\{\left|\frac{\partial J_N\left(\theta^0\right)}{\partial \theta_k}\right| < \tau_2\right\} + \frac{1}{h} \cdot M \cdot \sum_{k=1}^{m} \mathbb{I}\left\{\left|\frac{\partial J_N\left(\theta^0\right)}{\partial \theta_k}\right| \geq \tau_2\right\} \\
&= \frac{1}{h} \cdot \tau_2 \cdot \sum_{k=1}^{m} \mathbb{I}\left\{\left|\frac{\partial J_N\left(\theta^0\right)}{\partial \theta_k}\right| < \tau_2\right\} + \frac{1}{h} \cdot M \cdot \left(m - \sum_{k=1}^{m} \mathbb{I}\left\{\left|\frac{\partial J_N\left(\theta^0\right)}{\partial \theta_k}\right| < \tau_2\right\}\right) \\
&= \frac{1}{h} \cdot \left(M \cdot m - (M - \tau_2)\sum_{k=1}^{m} \mathbb{I}\left\{\left|\frac{\partial J_N\left(\theta^0\right)}{\partial \theta_k}\right| < \tau_2\right\}\right).
\end{aligned}
\tag{32}
$$

where $M \triangleq \max\left\{\left|\frac{\partial J_N\left(\theta^0\right)}{\partial \theta_k}\right| \,\middle|\, \left|\frac{\partial J_N\left(\theta^0\right)}{\partial \theta_k}\right| \geq \tau_2\right\} > \tau_2 > \tau_1 > 0.$

# B SWEET GRADIENT PHENOMENON

Sweet Gradient widely exists across different search spaces, datasets, the number of architectures, batch size, and initializations. Here we list the heatmaps of Sweet Gradient under various configurations.

## B.1 DIFFERENT SEARCH SPACES

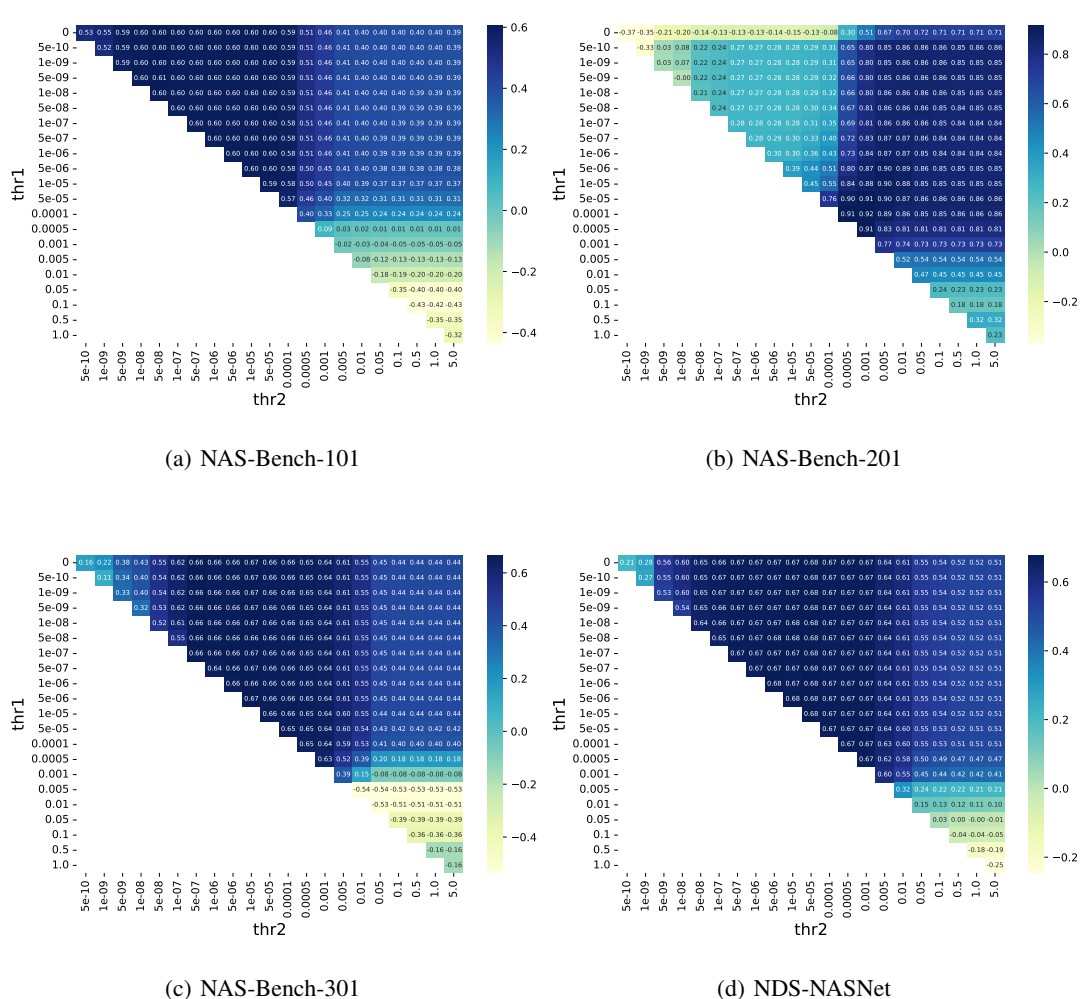

(a) NAS-Bench-101

(b) NAS-Bench-201

(c) NAS-Bench-301

(d) NDS-NASNet

Figure 4: Sweet Gradient across different search spaces.

## B.2 DIFFERENT DATASETS

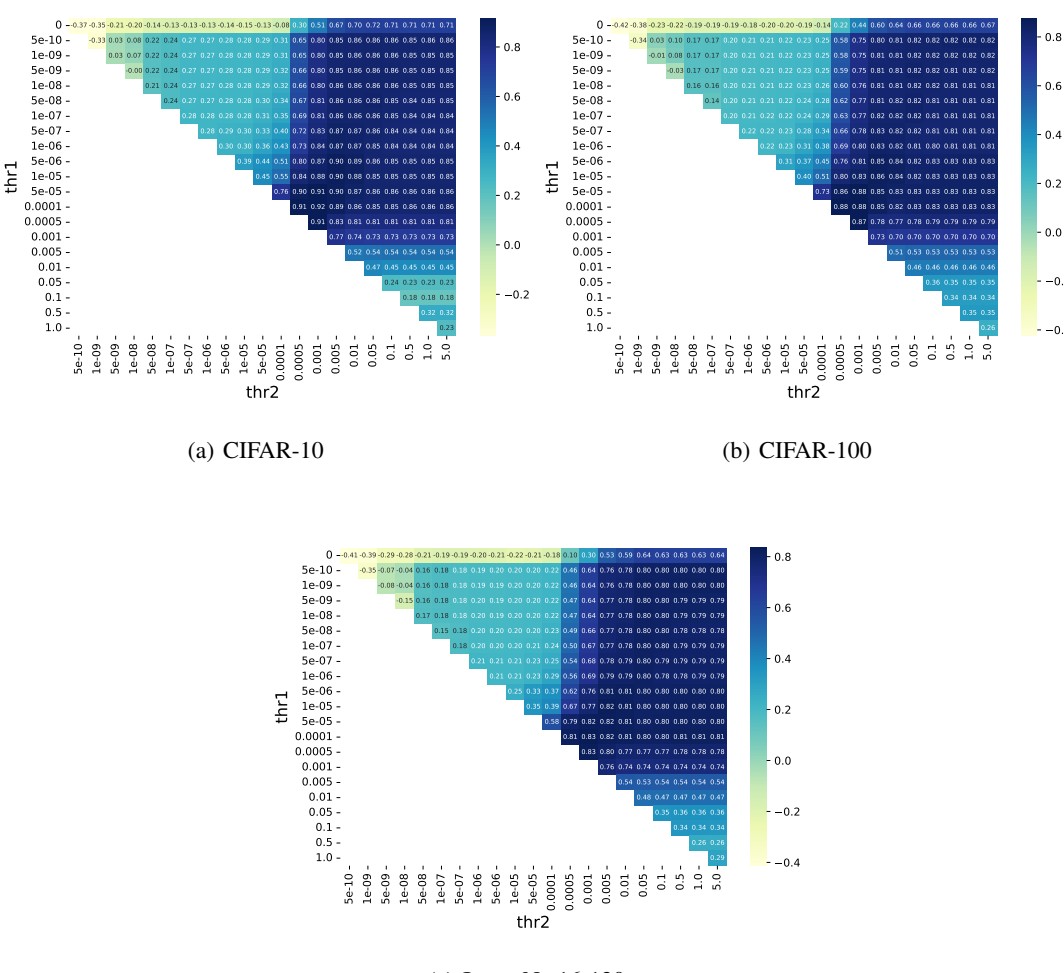

(a) CIFAR-10

(b) CIFAR-100

(c) ImageNet16-120

Figure 5: Sweet Gradient across different dataset in NAS-Bench-201.

## B.3 DIFFERENT NUMBER OF ARCHITECTURES

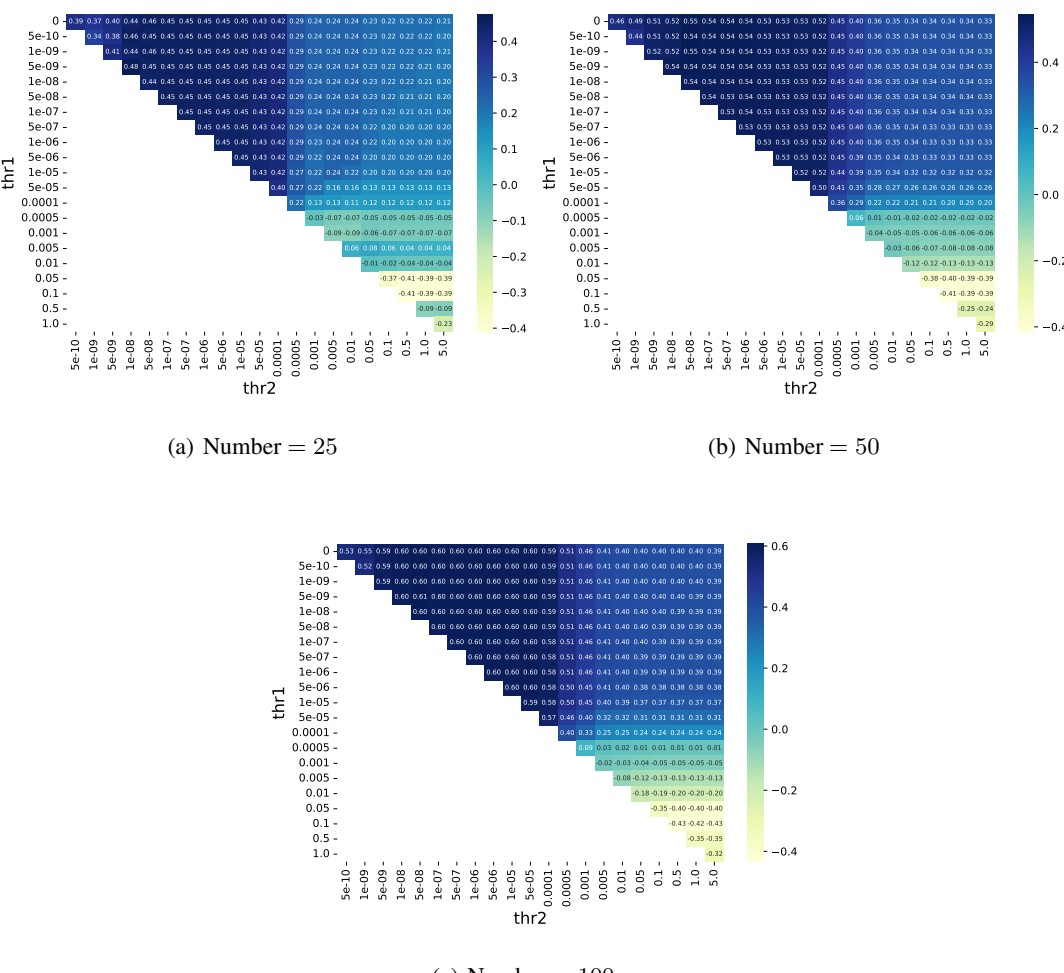

(a) Number = 25

(b) Number = 50

(c) Number = 100

Figure 6: Sweet Gradient across different number of architectures in NAS-Bench-101.

## B.4 DIFFERENT BATCH SIZES

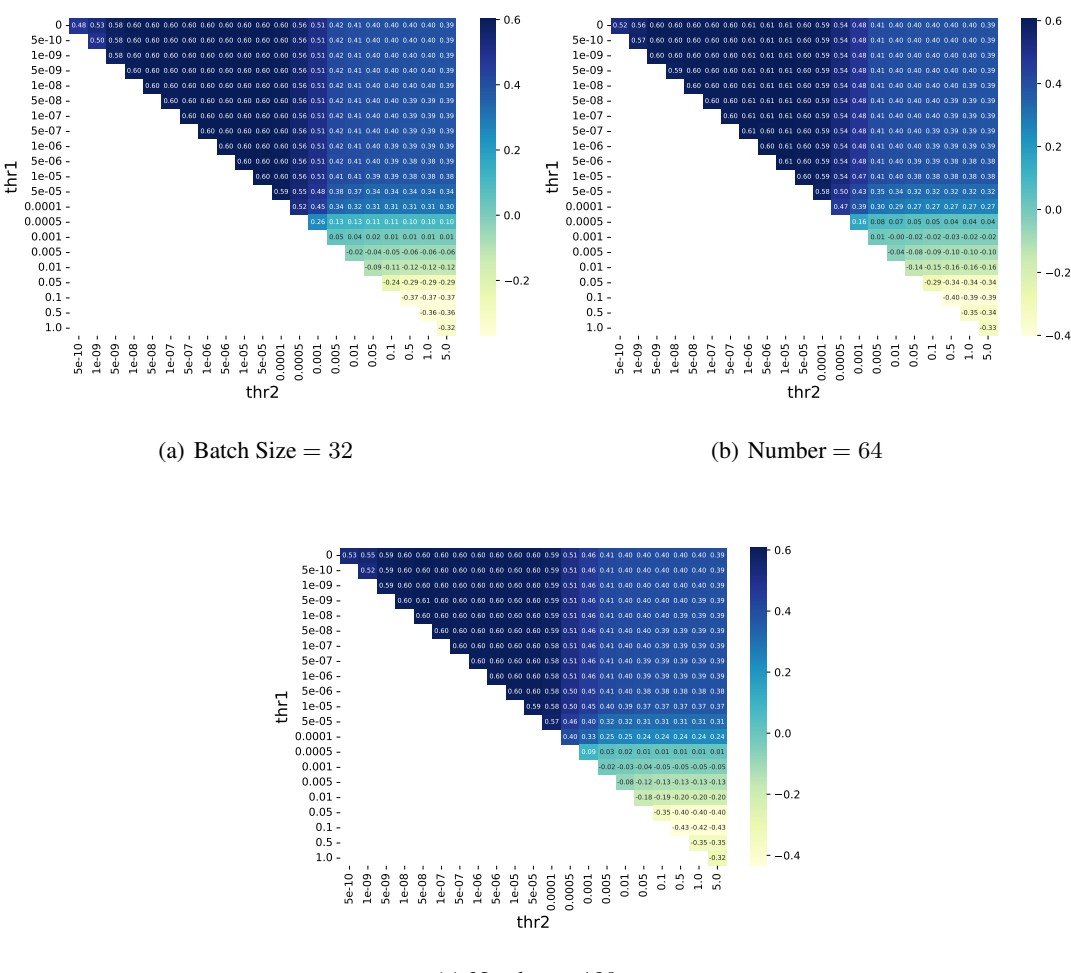

(a) Batch Size = 32

(b) Number = 64

(c) Number = 128

Figure 7: Sweet Gradient across different batch size in NAS-Bench-101.

## B.5 DIFFERENT INITIALIZATION

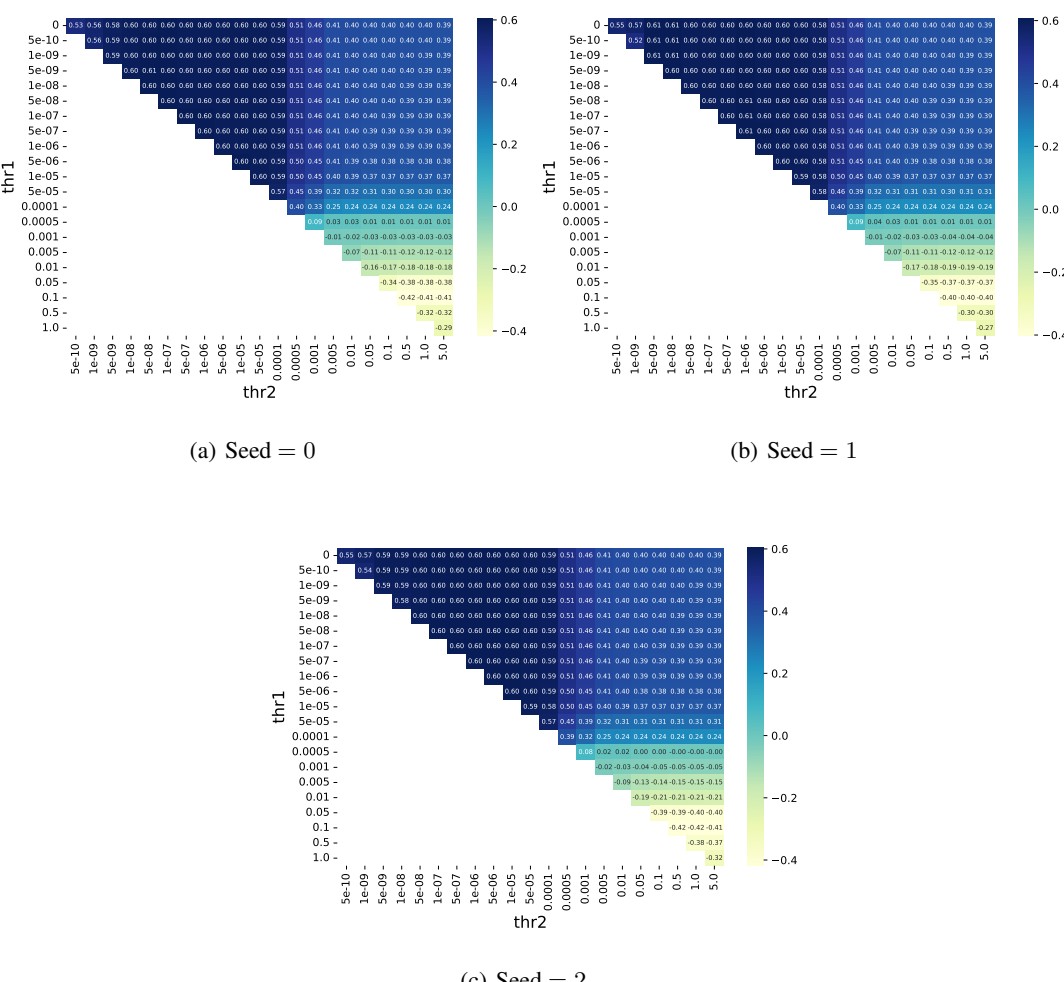

(a) Seed = 0

(b) Seed = 1

(c) Seed = 2

Figure 8: Sweet Gradient across different initialization in NAS-Bench-101.

# C   MORE INTERVALS IN NAS-BENCH-101 AND NAS-BENCH-201

## C.1   INTERVALS IN NAS-BENCH-101

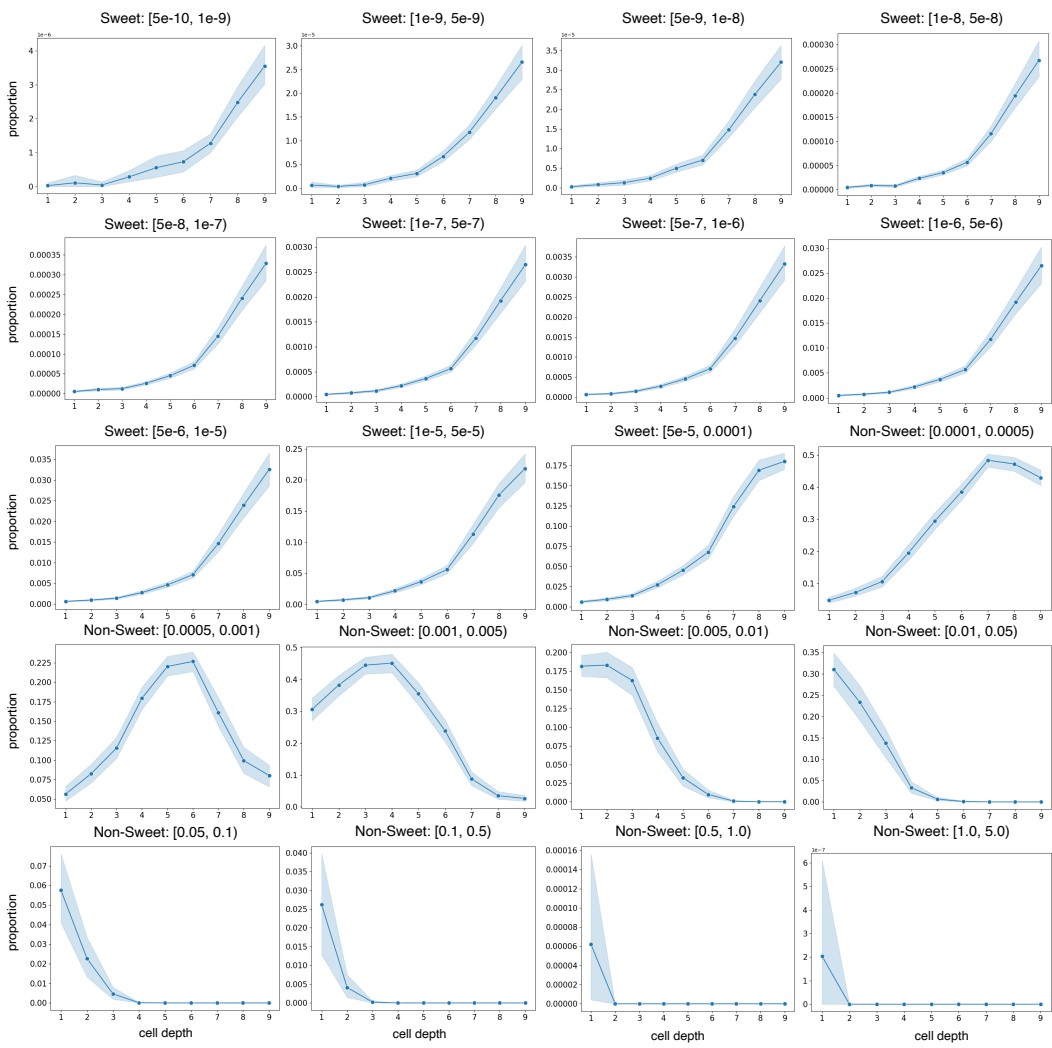

Figure 9: More Intervals in NAS-Bench-101.

## C.2 INTERVALS IN NAS-BENCH-201

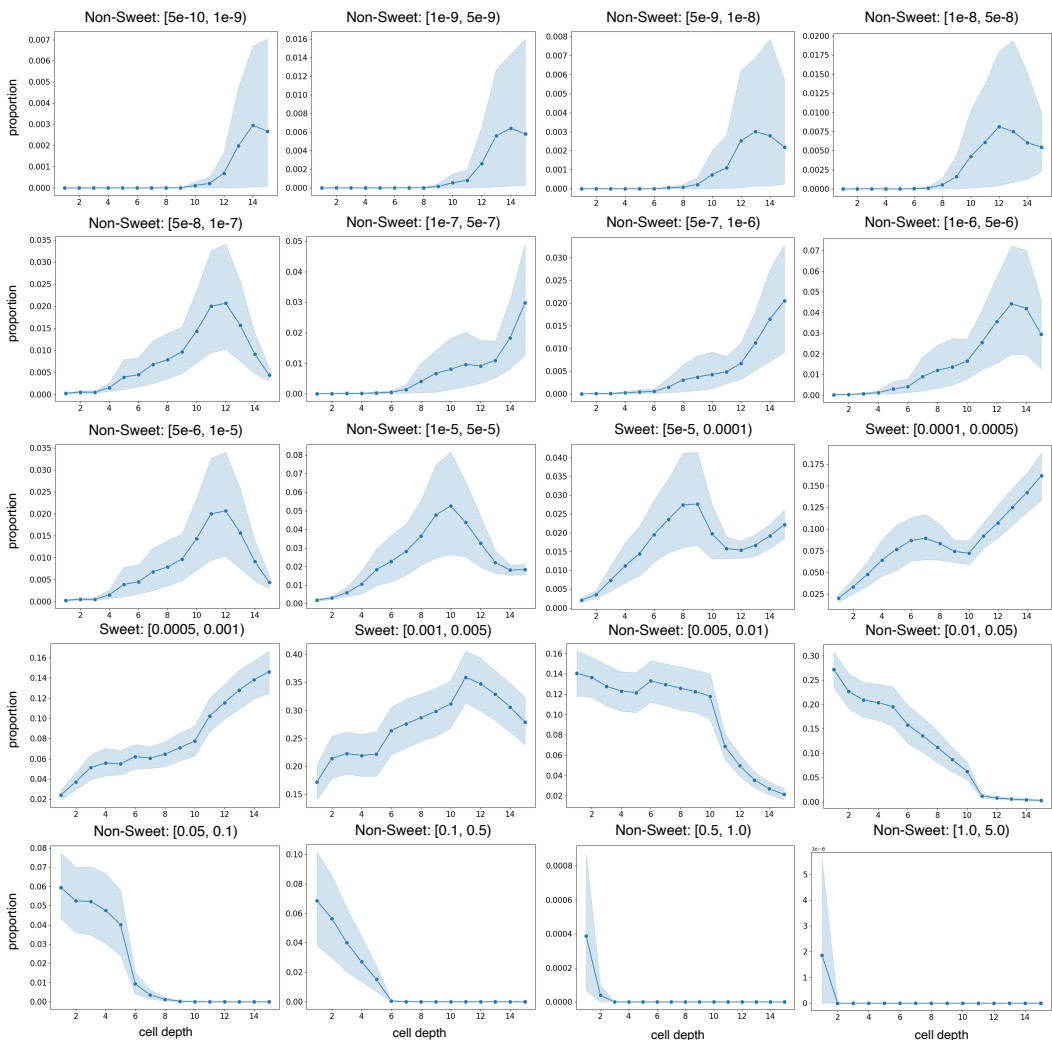

Figure 10: More Intervals in NAS-Bench-201.

# D    OTHER INVESTIGATIONS

**Investigation 1: Is the Sweet Gradient related to gradient distribution**? A natural conjecture is that Sweet Gradient is more likely to exist in intervals with concentrated gradient values. Figure 11(a) and 11(b) show the distribution of the absolute gradients values in NAS-Bench-101 and NAS-Bench-201, which disproves the conjecture. One reason is that there are Non-Sweet Gradient intervals where the gradients are concentrated, e.g., $[0.001, 0.005)$ in NAS-Bench-201. Another reason is that there are also Sweet Gradient intervals where the gradients are not concentrated, e.g., $[5e{-}7, 1e{-}6)$ in NAS-Bench-101.

**Investigation 2: Is the Sweet Gradient related to activation distribution**? The activation reflects the expressibility (Chen et al., 2021a) of the network, so does the activation value distribution affect the sweet gradient? Figure 11(c) and 11(d) show the distribution of absolute activation values in NAS-Bench-101 and NAS-Bench-201. The activation values are mainly distributed in $[0.01, 5)$, and the corresponding interval is almost staggered with Sweet Gradient with a large order of magnitude difference, which reveals that they are not related.

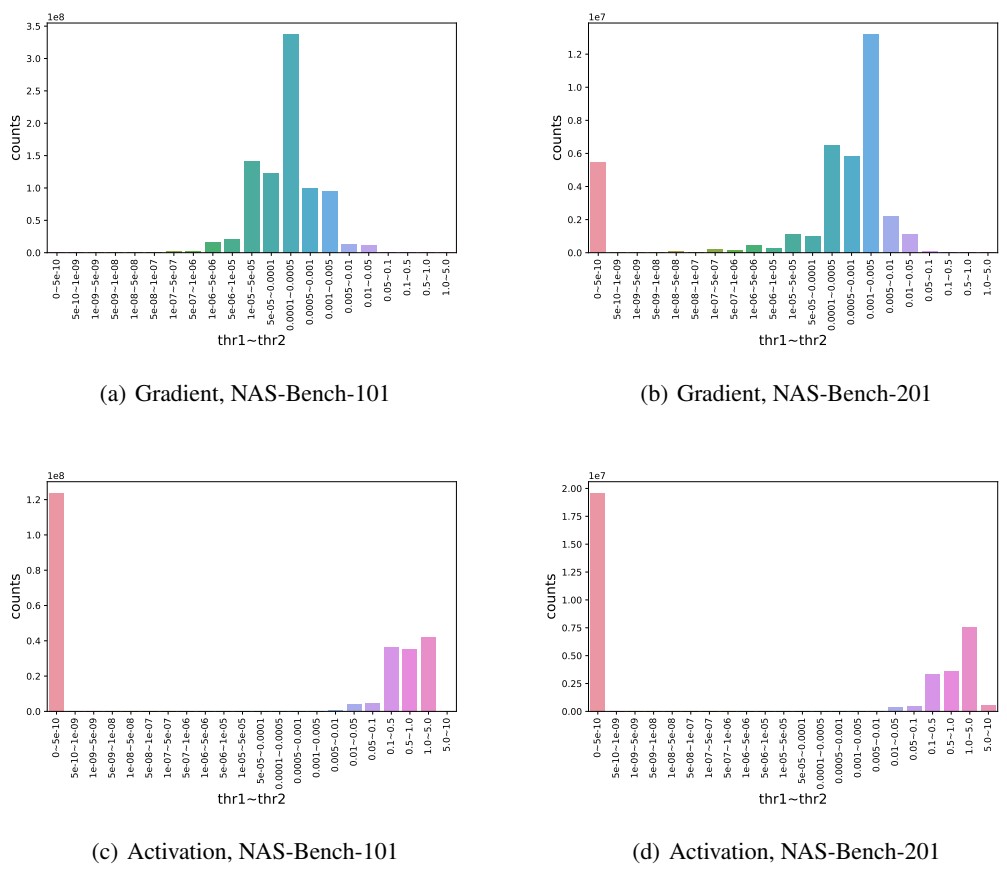

(a) Gradient, NAS-Bench-101          (b) Gradient, NAS-Bench-201

(c) Activation, NAS-Bench-101          (d) Activation, NAS-Bench-201

Figure 11: The gradient and activation distribution under different intervals in NAS-Bench-101 and NAS-Bench-201 (CIFAR-10).

# E    EXPERIMENTAL DETAILS

## E.1    DETAILS OF CONSISTENCY EXPERIMENTS

Experiments are mainly based on the publicly available codes from Mellor et al. (2021); Abdelfattah et al. (2021); Zhang & Jia (2022); Chen et al. (2021a); Lin et al. (2021); Mehta et al. (2022) (MIT or Apache-2.0 licence). Following Zhang & Jia (2022), the seed is 42 which is randomly chosen. The batch size is 64 in NAS-Bench-101, NAS-Bench-201, and NAS-Bench-301, and 128 in NDS space, which is also the same as Zhang & Jia (2022). For Sweetimator, we set the architecture number to 100, and the split to 2.

## E.2    DETAILS OF NAS-BENCH-201 EXPERIMENTS

Sweetimator has a batch size of 128, architecture numbers of 100, and a split of 2 in architecture selection and estimator-assisted NAS. For a fair comparison, we strictly followed the algorithm proposed in Mellor et al. (2021); Zhang & Jia (2022).

## E.3    DETAILS OF DARTS EXPERIMENTS

In the search phase, Sweetimator is obtained by a batch size of 128, architecture numbers of 100, and a split of 2 on CIFAR-10 and ImageNet. Then we integrate Sweetimator into REA algorithm by replacing training networks with computing the score of Sweetimator. Following Dong & Yang (2020), we set the number of cycles to 200, the population size to 10 and the sample size to 3. Although the cycle is only 200, the scores converge rapidly and the search results can be competitive with other Zero-Shot methods, which also bring a fast search speed. Therefore, we did not do more hyper-parameter tuning. Please note that the search time includes computing the best interval and searching for the optimal architecture.

In the retraining phase, we follow DARTS settings to train architectures. On CIFAR-10, the network consists of 20 layers with 36 initial channels. We utilized the SGD optimizer to train the network for 600 epochs with a batch size of 96. The learning rate decays from 0.025 to 0 by the cosine scheduler. Other settings like cutout, auxiliary, and path dropout are the same as DARTS. On ImageNet, the network consists of 14 cells with 48 channels, which is restricted to be less than 600M FLOPs. The SGD optimizer is used to train the network with 250 epochs, a learning rate of 0.5, weight decay of $3e-5$, and a batch size of 1024. We train the network on eight NVIDIA V100 for around three days.

## F    COMPARISON WITH NON-ZERO-SHOT ESTIMATORS

**Settings**. We compare Sweetimator with Non-Zero-Shot estimators in NAS-Bench-201. The baselines include SPOS (Guo et al., 2020), Neural Predictor (Wen et al., 2020), NAO (Luo et al., 2018), TNASP (Lu et al., 2021). For SPOS, we trained a supernet with 250 epochs and a batch size of 256, then utilized the validation accuracy of sub-networks as the estimator. For Neural Predictor, NAO and TNASP, we directly used the results in Lu et al. (2021). For Sweetimator, we set the batch size to 128, the architecture number to 100, and the split to 2. The compared metric is Kendall's Tau between the score of estimators and the test accuracy on CIFAR-10 of the benchmark.

**Results**. Table 7 shows the comparison results. Sweetimator can achieve superior rank consistency to Non-Zero-Shot estimators on NAS-Bench-201, which further demonstrate the effectiveness of the proposed method. Moreover, Sweetimator takes only a few minutes to obtain the best interval, which is more efficient than the process of training networks in Non-Zero-Shot estimators.

Table 6: Rank Consistency of Non-Zero-Shot estimators and Sweetimator in NAS-Bench-201.

| Estimators | Kendall's Tau | Type |
|---|---|---|
| SPOS | 0.621 | Non-Zero-Shot |
| Neural Predictor | 0.646 | Non-Zero-Shot |
| NAO | 0.526 | Non-Zero-Shot |
| TNASP | 0.724 | Non-Zero-Shot |
| Sweetimator | 0.736 | Zero-Shot |

# G VISUALIZATION OF SEARCH RESULTS

## G.1 NAS-BENCH-201

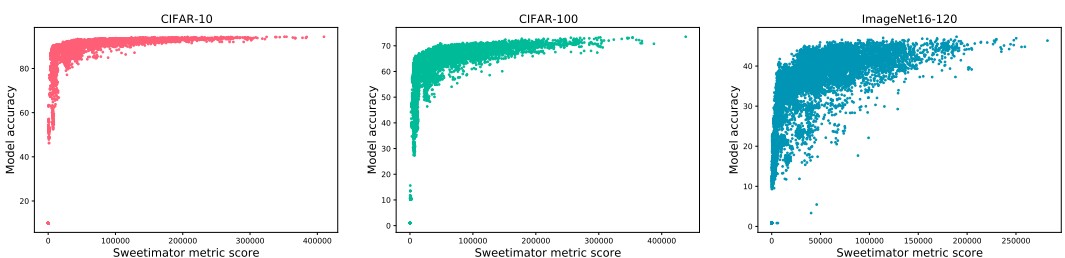

Figure 12: Visualization of model test accuracy versus Sweetimator metric score on CIFAR-10, CIFAR-100, ImageNet16-120.

## G.2 DARTS CELLS

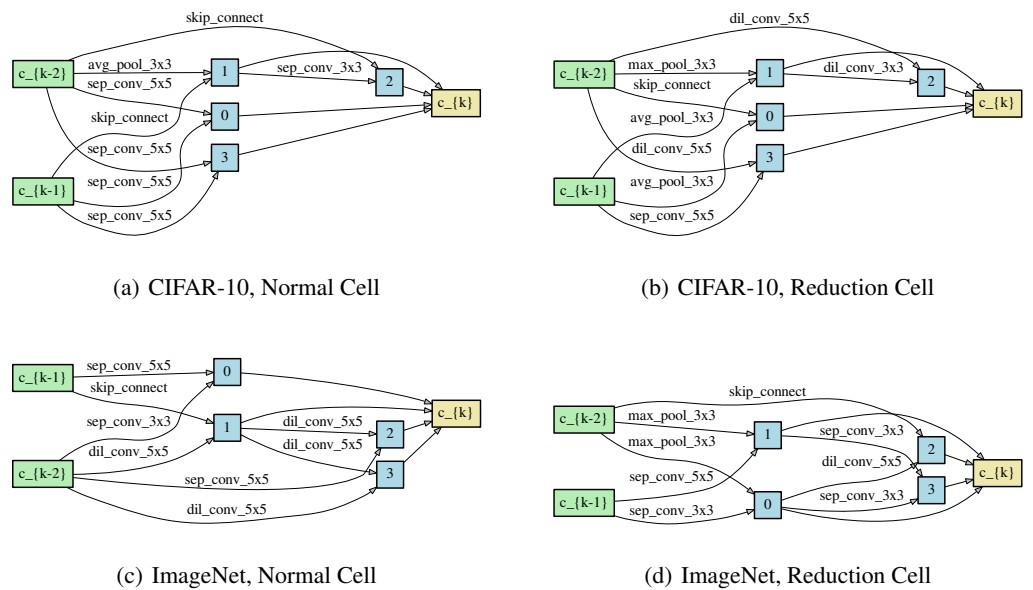

(a) CIFAR-10, Normal Cell

(b) CIFAR-10, Reduction Cell

(c) ImageNet, Normal Cell

(d) ImageNet, Reduction Cell

Figure 13: Visualization of the best searched cells on CIFAR-10 and ImageNet.

# H   MORE DIFFERENT TASKS

We conduct experiments on NAS-Bench-NLP and NAS-Bench-ASR benchmarks to further validate our method. The compared baselines include SynFlow, GradSign, and Parameters. The results in the following table show that Sweetimator has superior performance consistency than other methods. However, the Spearman's rank of NAS-Bench-NLP drops significantly compared to that of CV tasks. We conjecture that the text generation tasks and RNN architectures are more complicated than the classification tasks and CNN architectures, and furthermore, the hyper-parameters of the architecture candidates in the NAS-Bench-NLP search space are not well-tuned. Due to the limited time available for rebuttal, we will improve the consistency of Sweetimator for more tasks in the future.

Table 7: Rank Consistency of Non-Zero-Shot estimators and Sweetimator and other Zero-Shot estimators by Spearman's rank in NAS-Bench-NLP and NAS-Bench-ASR.

| Benchmark | SynFlow | GradSign | Parameters | Sweetimator |
|---|---|---|---|---|
| NAS-Bench-NLP | -0.062 | 0.134 | -0.046 | 0.159 |
| NAS-Bench-ASR | 0.244 | 0.288 | 0.305 | 0.311 |

