# OpenReview forum: "Sweet Gradient Matters: Designing Consistent and Efficient Estimator for Zero-Shot Neural Architecture Search"
_ICLR.cc/2023/Conference — Submitted to ICLR 2023_

### Official Review · Reviewer_FxrW · 2022-10-20

**Confidence:** 4
**Correctness:** 2
**Technical Novelty And Significance:** 3
**Empirical Novelty And Significance:** 2
**Recommendation:** 5

**Clarity, Quality, Novelty And Reproducibility:**

Overall, the clarity of this paper may be further improved (see the weakness above). Meanwhile, this paper should provide the reproducibility statement and source codes to support the reproducibility of its zero-shot estimator and NAS algorithm.

**Strength And Weaknesses:**

## Strength
1. This paper studies an essential problem in the zero-shot/training-free NAS field, i.e., improving the practical consistency of zero-shot estimators, and develops an interesting zero-shot estimator Sweetimator to tackle it, which empirically outperforms other zero-shot estimators in extensive experiments.
2. Based on the interesting Sweetimator, this paper proposes a zero-shot NAS algorithm that performs better than other zero-shot baselines on various NAS benchmarks and datasets.
3. This paper provides many ablation studies to help better understand their proposed Sweet Gradient interval.

## Weakness
1. The theoretical motivation or justification for the study of eq.1 is missing in this paper. Such a motivation or justification is necessary because it helps to understand why studying this eq.1 (rather than other forms of gradients) may help develop a consistent zero-shot estimator for NAS and how it helps improve the rank consistency over different NAS benchmarks and datasets. The whole paper only relies on empirical observations to support its conclusions, which is not so convincing to me.
2. This paper doesn't provide a formal, clear, and effective definition of Sweet Gradient intervals, which leads to some problems. Specifically, in this paper, it's informally defined by comparing with the rank consistency of #params in Sec. 3.2. This means that to obtain these Sweet Gradient intervals, people need to measure the rank correlations of #params on various NAS benchmarks, which seems to contradict with eq.3 that only leverages the target NAS benchmark to obtain Sweet Gradient intervals (Problem 1). In addition, based on such an informal definition, Sweet Gradient intervals adapt when the rank correlation of #params changes. So, the eq.1 based on Sweet Gradient intervals may not be able to outperform other zero-shot estimators when the rank correlation of #params performs very poorly in certain tasks (since Sweet Gradient intervals only need to improve over #params even slightly) and thus can not guarantee to achieve better consistency than other zero-shot estimators (Problem 2).
3. The effectiveness of using eq.3 to determine the best Sweet Gradient interval remains questionable. This is because (a) the discussion below eq.2 shows that the Sweet Gradient interval shifts with the increasing network depth and therefore the Sweet Gradient intervals obtained using the increasing network depth may differ from the desired one for the search space with a different fixed network depth (this is known to be common in NAS by searching with a smaller network depth and applying with a larger network depth), and (b) the parameter proportion in the Non-Sweet Gradient intervals may also increase according to Appendix B.2 and therefore eq.3 may not really output the best Sweet Gradient interval (it can also output Non-Sweet Gradient interval). This begs the question: Is the Sweet Gradient interval really the one that leads to the good search results in the experiments of this paper?
4. The details of Fig. 3 need to be clearer, e.g., the rank correlations of these Sweet Gradient intervals and which layer is used to plot (since eq.2 is defined on a specific layer l). These are important to understand the difference among Sweet Gradient intervals and the possible impacts of different layers on the observations in the paper.
5. As mentioned in the limitation section of this paper, this paper only focuses on classification tasks with different search spaces to compare the consistency of different zero-shot estimators. I think it may also be important to examine the improved consistency of the proposed method in different tasks to further support its superiority.

**Summary Of The Paper:**

While existing zero-shot NAS algorithms can usually achieve competitive performance in practice, their zero-shot estimators are unfortunately observed to perform inconsistently in different tasks (e.g., search spaces and datasets), which is even not comparable to the most straightforward one, i.e., the number of parameters, abbreviated to #params. This is known to be a critical problem in the zero-shot/training-free NAS field. So, this paper aims to develop a new zero-shot estimator that can perform consistently well on different benchmark datasets and search spaces. Before realizing this, this paper firstly observes that the number of Sweet Gradient (i.e., gradient within a special interval) consistently has better rank correlations than other zero-shot estimators in practice, which indicates that the number of Sweet Gradient may be a better choice for zero-shot NAS. Then, inspired by the close connection between the increasing proportion of Sweet Gradient and the increasing neural network depth, this paper proposes to leverage network depth to compute the Sweetimator and hence can find Sweet Gradient intervals without training. Finally, this paper uses extensive experiments to show that its zero-shot NAS based on Sweetimator can outperform existing zero-shot estimators in both ranking correlations and search results on various NAS benchmarks and datasets.

**Summary Of The Review:**

In general, my major concerns lie in the soundness and clarity of this paper as the empirical results are already quite good in this paper. I hope the authors can address my concern in the rebuttal period.

---

### Official Review · Reviewer_tGTD · 2022-10-24

**Confidence:** 4
**Clarity, Quality, Novelty And Reproducibility:** This paper has strong motivations and…
**Correctness:** 3
**Technical Novelty And Significance:** 3
**Empirical Novelty And Significance:** 3
**Recommendation:** 5

**Strength And Weaknesses:**

Strengths
1. The paper is very well-written and easy to follow.
2. The proposed idea to search “sweet gradients” in candidate networks seems novel and interesting.
3. The paper has provided extensive evaluations over different metrics and datasets. The results are presented in clear manner and are fairly convincing.

Weakness
1. The theoretical novelty is limited, although I do acknowledge the contribution of the proposed metric to zero-shot NAS.
2. The proposed method in this paper looks like a universal evaluation metric, not limited to the computer vision field. So the diversity of experiments is not enough. The authors only use image datasets to validate the effectiveness of the proposed metric. It is better to validate the proposed metric on NAS-Bench-NLP and NAS-Beach-ASR should


**Summary Of The Paper:**

The authors propose a zero-cost evaluation metric to improve zero-shot neural architecture search efficiency in this work. This work uses gradient distribution and network information at initialization for scoring candidate models and achieves a strong rank consistency with the performance of candidate models. Extensive experiment results are provided to validate the efficiency of the zero-cost metric.

**Summary Of The Review:**

This paper is well written and technically sound, but the theoretical novelty is limited.

---

### Official Review · Reviewer_3NxU · 2022-10-24

**Confidence:** 4
**Clarity, Quality, Novelty And Reproducibility:** 1. Clarity
**Correctness:** 3
**Technical Novelty And Significance:** 2
**Empirical Novelty And Significance:** 2
**Recommendation:** 5

**Details Of Ethics Concerns:**

There are no Ethics Concerns.

**Strength And Weaknesses:**

Strength:
1. The writing of the paper is clear and understandable.
2. The comparison between Sweet Gradient and other indicators is completely sufficient.

Weaknesses:
1. There is a lack of theoretical analysis on why Sweet Gradient is better than other measurements.
2. Lack of innovation in improving the Zero-shot NAS search method.

Question:
1. Figure 1 compares the results on three datasets. Has the author analyzed why there are gaps between the three datasets? If the differences between the three datasets are all very large. How does the author guarantee the versatility of Sweet Gradient?

2. Does the author theoretically analyze why Sweet Gradient is better than other indicators?

3. Is the improvement to the existing Zero shot NAS just updated with new metrics? I have a question about how the paper implements zero-shot NAS.

**Summary Of The Paper:**

Verifying Sweet Gradient: The paper first discovered through experiments that the Sweet Gradient of parameters, i.e., the absolute gradient values within a certain interval, brings higher consistency in network performance than the overall number of parameters. Then this paper also uncovered a positive correlation between the network depth and the proportion of parameters with sweet gradients in each layer.

NAS: Based on the observation, this paper also proposes a Zero shot NAS method based on Sweet Gradient.

**Summary Of The Review:**

The motivation for the research is clear. But the paper lacks the necessary theoretical analysis. I need the author to provide more details in the rebuttal to improve my score.

---

### Official Review · Reviewer_g7DU · 2022-10-25

**Confidence:** 4
**Clarity, Quality, Novelty And Reproducibility:** The paper is well-written, and the pr…
**Correctness:** 3
**Technical Novelty And Significance:** 3
**Empirical Novelty And Significance:** 3
**Recommendation:** 5

**Strength And Weaknesses:**

Strength:

1. Utilizing the absolute gradient values within a certain interval as zero cost proxy evaluating candidate models in NAS is novel and interesting. Building a consistent zero-cost proxy is a very good contribution.
2. The method is simple and easy to implement.
3. The paper is clearly written. Experiments and ablation studies are well performed.

Weakness:

1. The proposed method, in its current form, is mostly heuristic. It is still not clear how the authors find that Sweet Gradient of parameters can be good proxies and why it works consistently well. More insightful discussion or theoretical analysis may be required to fully justify the proposed method.


**Summary Of The Paper:**

This paper finds that the absolute gradient values within a certain interval, called Sweet Gradient of parameters can be good zero-cost proxy evaluating candidate models in zero shot NAS. The authors find a positive correlation between the depth and the proportion of sweet gradient parameters, and further propose a way to automatically determine the Sweet Gradient interval. The effectiveness of the proposed method is validated on four benchmarks with eight search spaces.

**Summary Of The Review:**

The paper is clearly written. The main concern is about insightful discussion and theoretical analysis.

---

### Decision · Program_Chairs · 2023-01-20

**Decision:**

Reject

**Justification For Why Not Higher Score:**

See the aforementioned major concern (A) and concern (B).

**Justification For Why Not Lower Score:**

N/A.

**Metareview: Summary, Strengths And Weaknesses:**

The main contribution of this work lies in proposing a zero-shot NAS estimator by exploiting the idea that the number of Sweet Gradient (i.e., the absolute gradient values within a certain interval) has better rank correlations than other zero-shot estimators empirically.

After reviewing and responding to the authors' rebuttal, the reviewers agree that the proposed idea is new and interesting, the paper is generally well-written and easy to follow, and the empirical evaluation is sufficiently extensive.

Initially, a major concern raised by all reviewers is the lack of theoretical analysis of the performance of the proposed zero-shot NAS estimator, among others. The authors have subsequently provided a theoretical analysis in their revised paper (Appendix A) and also clarified the assumptions for the analysis in their response to Reviewer FxrW.

(A) However, the advantages of using Sweet Gradient Interval over existing training-free metrics are still not sufficiently justified by the additional theoretical explanations.

(B) Furthermore, the authors have made a major revision to their manuscript in response to the reviewers' feedback and provided substantial clarifications even after that. We all agree that the future revised version will need another thorough, careful review and hence, it is more appropriate to be resubmitted to a future ML-related venue.